



Atmospheric
Chemistry
and Physics

# Source apportionment of atmospheric aerosol in a marine dusty environment by ionic/composition mass balance (IMB)

**João Cardoso[1,2], Susana M. Almeida[3], Teresa Nunes[1], Marina Almeida-Silva[3], Mário Cerqueira[1], Célia Alves[1], Fernando Rocha[4], Paula Chaves[3], Miguel Reis[3], Pedro Salvador[5], Begoña Artiñano[5], and Casimiro Pio[1]**

[1]CESAM & Dep. Environ, Aveiro University, Aveiro, Portugal
[2]Cape Verde University, Praia, Cabo Verde TS1
[3]TN, Instituto Superior Técnico, Lisbon University, Bobadela, Portugal
[4]Geobiotec & Dep. Geosciences, Aveiro University, Aveiro, Portugal
[5]Environ Dep, CIEMAT, Madrid, Spain

**Correspondence:** Casimiro Pio (casimiro@ua.pt)

**Abstract.** TS2 TS3 PM$_{10}$ aerosol was sampled in Santiago, the largest island of Cabo Verde, for 1 year, and analysed for elements, ions and carbonaceous material. Very high levels of dust were measured during the winter months, as a result of the direct transport of dust plumes from the African continent. Ionic and mass balances (IMBs) were applied to the analysed compounds, permitting the determination of six to seven different processes and source contributions to the aerosol loading: insoluble and soluble dust, sea salt, carbonaceous material and secondary inorganic compounds resulting from the reaction of acidic precursors with ammonia, sea salt and dust. The mass balance could be closed by the consideration and estimation of sorbed water that constituted 20 %–30 % of the aerosol mass. The balance methodology was compared with positive matrix factorisation (PMF), showing similar qualitative source composition. In quantitative terms, while for soil dust and secondary inorganic compound source classes, the results are similar, for other sources such as sea-salt spray there are significant differences in periods of dust episodes. The discrepancies between both approaches are interpreted based on calculated source profiles. The joint utilisation of the two methodologies, which are complementary, gives confidence in our capability for the correct source apportionment of aerosol particles.

## 1 Introduction

Atmospheric aerosol has an important role in atmospheric physics and chemistry (Lohmann and Feichter, 2005; Pöschl, 2005) and significant impacts on climate (Buseck and Pósfai, 1999; Ramanatham et al., 2001) and human health (Pope, 2000; Brunekreef and Fosberg, 2005; Tobias et al., 2011).

Atmospheric aerosol is both the result of gas-to-particle transformation from natural or anthropogenically induced precursors and of direct emissions from the Earth's surface by the action of wind on soil and water. Gas-to-particle transformation gives fundamentally fine (submicrometre) particles, while wind-induced mechanic processes on the planet's surface originate mostly from coarse particles as sea salt over the oceans or soil dust over deserts and other bare soil areas (Seinfeld and Pandis, 1998). On a global scale, sea spray and dust are highly dominant, in terms of suspended mass and regions affected, by comparison with other aerosol sources (Raes et al., 2000; Tanaka and Chiba, 2006). Although mostly natural, coarse aerosol particles associated with dust emissions are frequently affected by anthropogenic activities such as industry, transportation and agricultural practices at urban and regional/hemispheric scales (Almeida et al., 2006a; Ginoux et al., 2012).

It is important to know quantitatively the sources of atmospheric aerosols in order to correctly implement strategies and measures to control and reduce atmospheric particulate pollution and its effects on nature and humankind. There is an array of methodologies to evaluate the impact of sources

and precursors on atmospheric particulate loading that range from emission, dispersion and transport modelling, to source apportionment techniques (Blanchard, 1999). Source apportionment techniques use information on aerosol atmospheric composition and concentrations, at one or several locations, to infer quantitatively the sources and processes responsible for the particulate loading at the receptor (Almeida et al., 2006b; Belis et al., 2013).

In source apportionment, when the number and composition of the sources are unknown, multivariate analysis, based on particulate composition variability at the receptor(s) as a function of time, is a very common and useful methodology to quantitatively evaluate the main sources of atmospheric particulate matter (Ashbaugh et al., 1984; Henry et al., 2004 TS4; Hopke, 1985). Presently, the most used multivariate analysis methodology is positive matrix factorisation (PMF), because it allows the discrimination of only positive values in both source profiles and contributions (Paatero and Tapper, 1994; Paatero, 1999; Reff et al., 2007; Amato et al., 2016).

Source apportionment multivariate methodologies permit frequently to identify the impact of the majority of direct sources and gas-to-particle conversion processes and their variability in time, during the aerosol measured period. If associated with statistical backward trajectory analysis, the method also permits the determination of source regions during regional and, principally, long-range transport (Salvador et al., 2016).

Multivariate methods, although very useful, are not perfect and have uncertainties resulting from collinearity of sources, the evolution of composition during transport, etc. that need to be detected and estimated (Belis et al., 2013). When a large number of compounds and elements is determined by chemical analysis of aerosol samples, an alternative methodology can be used to infer the total aerosol composition, which takes into account that the total aerosol mass is the sum of the mass of the individual components. Also from the chemical analysis, it is possible to intercompare analysed anions and cations, which have to obey the principle of electroneutrality. From the mass and ionic composition, it is frequently viable to infer quantitatively the origin of the aerosol, because many of the analysed constituents are tracers of specific sources.

Ionic and mass balances (IMBs) rely mostly on the direct measurement of aerosol constituents and therefore are less affected by indemonstrable assumptions, as it happens in the assignment of the number of factors and their identities, in multivariate methods such as PMF (Belis et al., 2014). Mass balance has been frequently applied in the past but mostly as a screening tool (Watson et al., 2002). However, if properly applied, ionic and mass balances have the potential to correctly perform the source apportionment of atmospheric aerosol. We would like to emphasise that the European guide on air pollution source apportionment with receptor models (Belis et al., 2014) exhorts to use receptor models in combination with independent methodologies to achieve more robust estimations by mutual validation of the outputs. Our objective in this paper is thus to develop and apply a detailed ionic and mass balance to aerosol particles in a dusty marine environment, demonstrating the capability of this methodology to determine the aerosol sources with an accuracy as good as that of the most developed multivariate methods, such as PMF.

## 2   Mass balance methodologies

Composition and mass balance is feasible when the main components of the aerosol sources, such as soil elements, sea-salt constituents, inorganic water soluble ions and carbonaceous mass, are measured and quantified (Sciare et al., 2005; Guinot et al., 2007; Grigoratos et al., 2014; Genga et al., 2017). Even when there is a thorough quantification of aerosol constituents, it is not often possible to apportion more than 70 % to 80 % of measured aerosol total mass, because important elemental constituents of particulate matter, such as oxygen, in water, soil and organic carbon, are not usually analysed (Malm et al., 1994; Andrews et al., 2000; Rees et al., 2004; Perrino et al., 2013; Grigoratos et al., 2014). Oxygen is the most abundant element in the Earth's crust, constituting on average 47 % of the continental crust mass (Wedepohl, 1995).

For mass balance purposes, the determination of soil contribution is, usually, inferred from the analysis of the major soil elements measured in the aerosol samples (Si, Al, Fe, Mn, Ti, etc.), taking into account the presence of their oxides:

$$\text{Soil dust mass} = SiO_2 + Al_2O_3 + Fe_2O_3 + MnO$$
$$+ TiO_2 + \text{etc.} \tag{1}$$

Depending on the completeness of the soil elemental analysis and the composition knowledge of the source soils, it is possible to adapt the above equation to better apportion the soil mass by mass balance (Formenti et al., 2001; Eldred, 2003; Andrews et al., 2000, and references therein):

$$\text{Soil dust mass} = F(2.14Si + 1.89Al + 1.43Fe$$
$$+ 1.39Mn + 1.67Ti + \text{etc.}), \tag{2}$$

where $F$ is a multiplying factor that takes into account the unmeasured material (such as elements and the presence of hydrated water) in the soil dust.

The sea-salt contribution is evaluated by considering that emitted sea-salt spray has the composition of salt in seawater. A possible exception is chloride that frequently appears in lower ratios due to sea-salt spray interaction with atmospheric acids, such as $HNO_3$, $H_2SO_4$ or $SO_2$, which results in the evaporation of $Cl^-$, as HCl. The $Cl^-$ in the particulate phase can be, partially, or totally, substituted by $NO_3^-$, or

$SO_4^{2-}$, in the form of secondary sodium and magnesium nitrates and sulfates (Pio and Lopes, 1998). A similar reaction can take place between these atmospheric strong acids and dust, resulting, for example, in the partial or total vaporisation of carbonates, as $CO_2$, and the formation of secondary calcium nitrates and sulfates (Pio et al., 1994; Goodman et al., 2000).

Soil carbonates are part of the carbonaceous aerosol. As they are infrequently analysed, in source apportionment they have to be approximately inferred from calcium and magnesium measurements. In dusty environments, the measurement of carbonates is important to permit a more correct composition/mass balance source apportionment.

Another component of the carbonaceous aerosol is the organic mass. This component is usually calculated from the measurement of organic carbon by applying a multiplication factor to take into account other unmeasured elements such as nitrogen, sulfur and, principally, oxygen. Factors ranging from 1.2 to 2.3 have been employed for this purpose (Countess et al., 1980; Japar et al., 1984; Rogge et al., 1993a, b; Sempere and Kawamura, 1994; Russel, 2003; Chen and Yu, 2007; El-Zanan et al., 2009). The highest values are commonly used in sites affected by biomass burning emissions, rich in sugars and organic acids, or away from emission sources, because, under these conditions, the precursor organic material had plenty of time to be strongly oxidised (Turpin and Lim, 2001; Sciare et al., 2005; Ervens et al., 2011). Genga et al. (2017) used variable values between 1.8 and 2.1, depending on the direction of the wind, to best fit the mass closure process in a Mediterranean port city.

Water is a common and important component of atmospheric aerosol that may constitute up to 20 % of the total PM mass (Canepari et al., 2013; Perrino et al., 2013). Model calculations estimate that particle-bound water constitutes 20 %–35 % of the annual mean European atmospheric PM concentrations (Tsyro, 2005). In spite of that, only in a few studies has aerosol particulate water been indirectly or directly estimated (Dick et al., 2000; Rees et al., 2004; Stanier et al., 2004; Speer et al., 1997, 2003; Kitamori et al., 2009).

Several attempts have been made and published to account for water in sampled aerosols. Using a thermodynamic equilibrium ion modelling, temperature, humidity and inorganic ions concentrations, Chen et al. (2014) TS6 estimated that water constituted up to 38 % of the $PM_{2.5}$ mass in the heavily polluted atmosphere of Beijing for aerosols weighted at 40 % relative humidity (RH). To estimate strongly bound water, Harrison et al. (2003), in samples weighted at 45 %–50 % RH, applied a hydration multiplication factor of 1.29 to the measured sulfates and nitrates (as ammonium and/or sodium compounds). Sciare et al. (2005) and Genga et al. (2017) successfully used this methodology to close the mass balance in Mediterranean aerosols.

During laboratory studies with water and sea-salt particles, Tang et al. (1997) found the presence of a hysteresis supersaturation when decreasing relative humidity, with sudden ef-

florescence at 47 % RH. Depending on whether the particles were in a dry or wet state, the ratio of water to dry sea-salt masses observed at 50 % RH was 0.4, or 1.4, respectively.

Tang and Munkelwitz (1994) and Shu et al. (1998) TS7 determined the water content in ammonium sulfate. A water/salt ratio of 0.4 was obtained at 50 % RH in liquid metastate equilibrium. A ratio of 0.45 was employed to calculate the water contribution to ammonium sulfate aerosols by Speer et al. (2003).

Speer et al. (2003) also estimated the water content in organic aerosol particles. A relationship between the excess liquid water and the measured organic carbon mass was found. Through modelling it was determined that, on average, about 80 % of the liquid water in the $PM_{2.5}$ could be accounted for by inorganic ions, with the remaining 20 % associated with organic compounds. The liquid water to organic carbon mass ratio, at 50 %, was estimated as 0.2 (an OM/OC value of 2 was considered).

## 3 Experimental

The present work uses data from the field campaign of the CV-DUST (Atmospheric aerosol in the Cabo Verde region: carbon and soluble fractions of $PM_{10}$) project, which took place on Santiago island, Cabo Verde, between January 2011 and January 2012. Atmospheric aerosol and ancillary measurements were performed on the roof platform of the Cape Verde Meteorological Institute, on the outskirts of Praia (14.92° N, 23.48° W), 98 m a.s.l. and approximately 1.7 km from the sea border. During the sampling period, daily averaged temperatures and RH ranged from 21 to 29 °C and from 50 % to 86 %, respectively. Total rainfall was only 152 mm, concentrated in the months of August to October.

$PM_{10}$ aerosol particles were collected, in parallel, onto three filter types (quartz fibre, Teflon and Nuclepore membranes) with high-volume and low-volume samplers, equipped with $PM_{10}$ inlets. A total of 140 events were sampled, with filtering periods ranging from 6 to 96 h (the low sampling periods during Saharan dust episodes) allowing the collection of enough aerosol material for all necessary analysis without the risk of clogging the filters during dust storms. Taking into account the variable extension of sampling periods, in this publication, all the calculated concentration averages are weighted by the sampling time.

Details of sampling and filter analysis are given elsewhere (Almeida-Silva, 2014; Salvador et al., 2016); here, we only provide a summary of published information. Filters were weighted with semi-micro- or microbalances to determine $PM_{10}$ total mass, at 50 % RH and 20 °C. Mass concentrations measured in the three parallel sampling lines compared quite well ($R = 0.99$; best-fit lines with $y/x = 0.98 - 1.02$; for details, see Fig. S1 in the Supplement), a confirmation that the filters were sampling the same aerosol population.

Nuclepore filters were employed to determine particulate elemental content using particle-induced X-ray emission (PIXE) and/or $k_0$-instrumental neutron activation analysis ($k_0$-INAA) (Almeida et al., 2013; Almeida-Silva, 2014).
A total of 26 elements was measured by the two techniques, although some light elements, such as Na and Cl, could only be quantified with large uncertainties, characteristic of the analytical conditions and techniques.

The quartz filters were used to determine carbonates
by acid evolution and non-dispersive infrared analysis of evolved $CO_2$ (Pio et al., 1994) and elemental carbon (EC) plus organic carbon (OC), with a homemade thermo-optical carbon analyser, after pre-removal of carbonates with HCl fumes (Pio et al., 2011).
Water-soluble anions and cations, sampled in Teflon filters, were measured by ion chromatography. The method permitted the quantification of $NH_4^+$, $Na^+$, $Mg^{2+}$ and $K^+$ cations, and $SO_4^{2-}$, $NO_3^-$ and $Cl^-$ anions. Comparison between total cation and anion equivalents indicates a clear
excess of cations (42 %, on average). Inclusion of independently measured carbonates in the ionic balance brings the ratio of cations to anions to 0.93, demonstrating the importance of carbonate measurements for a more complete aerosol characterisation in dusty environments (see Fig. S2
in the Supplement for details).

## 4 Results and discussion

### 4.1 PM$_{10}$ mass and components

As reported elsewhere (Almeida-Silva, 2013; Pio et al., 2014; Salvador et al., 2016), Cabo Verde has two dis-
30 tinct atmospheric pollution seasons. During winter months (December–February) the atmospheric boundary layer is impacted by important dust intrusions from the Saharan region, with daily averaged PM$_{10}$ concentrations going up to hundreds of µg m$^{-3}$ (see Fig. 1 and Table 1). This period is lo-
35 cally designated as "Bruma-Seca", meaning "dry haze".

During May–September, there is no direct intrusion of dust plumes from Africa, at lower atmospheric levels, in the boundary layer (represented by a blue shade mask in Fig. 1), and we call it dust-free season. During the dust-free period,
the atmosphere contains still important quantities of dust originating either from the island arid surface or from continuous dust transport from Africa into the region across the free troposphere, which partially sediments to lower atmospheric levels (Gama et al., 2015). The months of March,
April, October and November have intermittent direct intrusions of Saharan dust, with PM$_{10}$ concentrations sometimes reaching 100 µg m$^{-3}$. Throughout this document, we present average results for the total sampling period and for the two dry-haze and dust-free seasons.

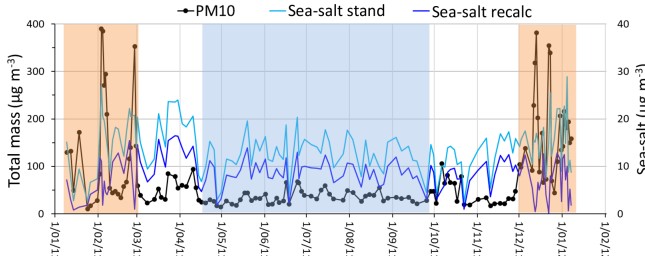

**Figure 1.** Levels of PM$_{10}$ and sea salt along the annual sampling period. Shades of brown and blue represent, respectively, periods clearly with (dry haze) and without (dust-free) dust plume direct intrusions. "Sea-salt stand" indicates concentrations of sea-salt spray calculated by considering Na$^+$ as an exclusive tracer of marine emissions. "Sea-salt recalc" represents sea-salt levels estimated after removal of soil dust Na$^+$ and Mg$^{2+}$.

**Table 1.** Weighted average concentrations of PM$_{10}$, major elements, ions and carbon fractions for the total campaign and for dry-haze and dust-free seasons. Na and Cl (in bold TS8) were measured with important inaccuracies, as evidenced by comparison with the respective water-soluble ion concentrations.

|  | Year average (ng m$^{-3}$) | Dry haze (ng m$^{-3}$) | Dust-free (ng m$^{-3}$) |
|---|---|---|---|
| PM$_{10}$ | 59 602 | 117 000 | 33 900 |
| Si | 6595 | 17 100 | 2150 |
| Al | 3814 | 9930 | 1220 |
| **Na** | **3230** | **3440** | **3040** |
| Fe | 1835 | 4560 | 721 |
| Ca | 1450 | 2920 | 783 |
| Mg | 969 | 2130 | 454 |
| K | 772 | 1580 | 380 |
| Ti | 197 | 454 | 93 |
| Ba | 31 | 66 | 21 |
| Mn | 31 | 78 | 12 |
| **Cl** | **4660** | **5810** | **3930** |
| S | 852 | 941 | 792 |
| Na$^+$ | 4047 | 4360 | 3760 |
| Ca$^{2+}$ | 818 | 1540 | 475 |
| Mg$^{2+}$ | 386 | 443 | 351 |
| K$^+$ | 240 | 251 | 166 |
| NH$_4^+$ | 213 | 101 | 257 |
| Cl$^-$ | 5344 | 5770 | 4840 |
| SO$_4^{2-}$ | 1898 | 1880 | 1880 |
| NO$_3^-$ | 1191 | 1250 | 115 |
| CO$_3^{2-}$ | 816 | 2230 | 169 |
| EC | 188 | 110 | 270 |
| OC | 980 | 1340 | 870 |

### 4.2 Soil and sea-salt sources

PM$_{10}$ in Cabo Verde is mainly influenced by emissions from soil and sea surfaces (Gama et al., 2015). The determina-

tion of the soil contribution, by composition/mass balance, can be improved from the knowledge of source soil composition. When information on dust-originated soil is unavailable, average global upper continental crust composition is frequently employed (Manson TS9 and Moore, 1982; Wedepohl, 1995). In our case, dust is almost exclusively originating from the north African Sahara and Sahel, and information on regional soil composition for these regions is available.

There is a handful of publications on soil composition in north Africa, which provide evidence of a wide composition variability across the Saharan and sub-Saharan regions (Guieu et al., 2002; Journet et al., 2013 TS10; Scheuvens et al., 2013, and references therein). One of the most complete Saharan soil data sets was given by the IDAEA-CSIC research group (Moreno et al., 2006). The publication provides the concentrations of 47 elements in the bulk soil of nine locations, across four regions, in north Africa (Hoggar massif, Chad Basin, Niger and western Sahara). Castillo et al. (2008) provides soil size distributed composition information for the same sites. We used this data set to infer the composition and mass balance of soil dust in our samples (Table S1, in the Supplement, adapted from Moreno et al., 2006, shows compound average contributions).

Saharan soil composition in Moreno et al. (2006) reveals some differences by comparison with the average crust/upper crust (Manson and Moore, 1986; Wedepohl, 1995), with a relative enrichment of Si and Ti, probably as a result of intense weathering of Sahara desert soils. Si and Ti form rather hard crystals (silica and rutile), resistant to physical weathering. The size chemical speciation of Saharan soils by Castillo et al. (2008) revealed Al, Mg and Fe moderate enrichments, in suspended finer particles, in contrast to K, which had increased concentrations at coarser sizes.

Aerosols generated by suspension from Sudan desert soil have also shown an Al enrichment, while there has been an impoverishment in Si and Ti elements in smaller particles (Eltayeb et al., 2001). The ratio of Al/Si in suspended dust decreased with increasing particle size. A similar behaviour occurred in the ratio of Al/Ti for particles $< 45\,\mu m$. The authors attributed this behaviour to the presence of large quartz crystals in soil and their substitution in dust by smaller particles of alkali–plagioclase and clay minerals.

Journet et al. (2014) concluded that, in desert soils, silica minerals accumulate preferably in the silt fraction ($2\,\mu m < D_p < 65\,\mu m$), while kaolinite and other clay minerals are mostly concentrated in the clay size fraction ($D_p < 2\,\mu m$); kaolinite that has a Al/Si ratio of 0.95 is the main mineral of desert areas. The authors assumed that the mineral composition of airborne dust is broadly similar to that of the clay size fraction in the desert soil.

Taking into account the previous information, we speculate that, as a result of soil weathering, particles containing Si are heavier/larger than other soil particles and therefore are more difficultly suspended by the wind and exported to other regions, enriching local soils.

Consequently, it is expectable that Saharan suspended dust will be impoverished in Si and Ti, by comparison with less hard minerals containing Al, Fe, Mg, Na, etc. This is observed in our aerosol samples where there is a quite constant Al/Si mass ratio of 0.61, independent from the period of the year or the air mass trajectories ($Al = 0.61Si - 254\,ng\,m^{-3}$; $R = 0.99$). Comparison between prevalently soil-originated elements, showed that, for Al, Fe and Mn, the concentration ratios in the aerosol are similar to those in average crustal material and within the limits of Saharan data from Moreno et al. (2006). In contrast, the ratio between particulate Al and Si (or Ti) levels is 2 to 4 times higher than that in Moreno et al. (2006), indicating a Si (Ti) deficit, by comparison with other major Saharan soil elements (for clarification, see Fig. S3 in the Supplement). After aerosol measurements performed in southern Morocco, Kandler et al. (2006) TS11 concluded that the major dust constituents were quartz, potassium feldspar, plagioclase, calcite, hematite and clay minerals. Particles in the range of 0.5–50 µm consisted mainly of silicates and, above 50 µm, quartz was dominant.

Published information concerning the Al/Si mass ratios in atmospheric dust from the Saharan region is still scarce. Al/Si average ratios of 0.43–0.49, with values, depending on the air mass origin, were measured by Chiapello et al. (1997), using bulk filtration, on Sal island, Cabo Verde. Formenti et al. (2003) determined Al/Si ratios of the order of 0.5, in particles larger than 1 µm, during aircraft measurements performed in the Cabo Verde region. Remoundaki et al. (2013) found Al/Si ratios of $0.44 \pm 0.12$ in $PM_{2.5}$ aerosols collected in Greece under the influence of air mass transport from the Saharan region. In South America, Formenti et al. (2001) observed Al/Si values of $0.48 \pm 0.08$ in fine particles and of $0.77 \pm 0.18$ in the coarse aerosol fraction. Aerosol measurements over the western Atlantic and the eastern coast of the US present an Al/Si ratio value coincident with our measurements (Eldred, 2003). From this information, we hypothesise that during long-range transport of Saharan dust there is a prevalent loss, by sedimentation (or non-emission), of Si (and Ti), in comparison to other particulate dust components, which becomes more evident for larger particles.

Because of the Al/Si behaviour in our samples, we felt more confident in using Fe as a representative tracer of soil contribution in composition/mass balance calculations. In addition, Fe is the soil element that showed a better correlation ($R = 0.99$) with $PM_{10}$ total mass during dust events (see Fig. S4 in the Supplement for visualisation).

In coastal, or marine, non-dusty environments, it is common, and correct, to infer the mass contribution of particulate sea spray to the atmospheric aerosol by using $Mg^{2+}$ or, predominantly, $Na^+$, as an exclusive sea-salt tracer. However, these ions are also present in the soil and, during dust episodes, the soil contribution cannot be neglected. From Fig. 1 it is possible to observe that sea-salt concentrations calculated in this way increase steeply during dust pollution episodes, which is not reasonable.

To eliminate the soil interference in sea-salt estimation, we employed $Fe/Mg^{2+}$ and $Fe/Na^+$ mass ratio superior edge lines (see Fig. 2). Similarly to Pio et al. (2011), edge lines' estimation is based on tracing a linear best-fit line through the 5 % of total concentration points (seven points in our case) with the highest ($Fe/(X - X_{min})$) ratios, where, presently, $X$ is $Na^+$ or $Mg^{2+}$ ion concentration and $X_{min}$ is the respective measured minimum ion concentration. These edge lines represent the minimum fractional contribution of sea salt to $Na^+$ and $Mg^{2+}$ in the aerosol, compatible with experimental data. Therefore, it is expectable that they represent, reasonably well, the ratios between the ions and Fe, in soil dust, as long as it is acceptable that, in such a location, these are the unique major sources of Fe and $Na^+$. Based on these edge line ratios, the amounts of soil $Mg^{2+}$ and $Na^+$ ions can thus be determined and subtracted from the total ion concentrations, permitting a first estimation of sea-salt $Mg_s^{2+}$ and $Na_s^+$. As these edge lines only approximatively represent the soil ratios, the calculation of sea-salt contributions may consequently suffer from these inaccuracies/variabilities.

A further refinement of sea-salt calculation can be implemented by looking at the ratios ($Na_s^+/Mg_s^{2+}$) calculated from $Na_s^+$ and $Mg_s^{2+}$ in each sample and comparing them with those in seawater ($Mg_{ss}^{2+}/Na_{ss}^+ = 0.12 \, \mu g \, \mu g^{-1}$, Turekian, 1968). Since it is not possible to have less $Mg_{ss}^{2+}$ (or more $Na_{ss}^+$) ion mass sea salt than the one given by the 0.12 ratio, if $Mg_s^{2+}/Na_s^+ \geq 0.12$, $Na_s^+$ is chosen as the true sea salt $Na_{ss}^+$ concentration. Otherwise, $Mg_s^{2+}$ is chosen as the true $Mg_{ss}^{2+}$ tracer. The contributions of other sea-salt ions are, consequently, estimated from the chosen $Na_{ss}^+$ or $Mg_{ss}^{2+}$, using the salt ratios present in seawater (Turekian, 1968).

Figure 1 presents the estimation of sea-salt contribution to the aerosol (Sea-salt recalc), considering the methodology described above. The figure shows that, with the modified methodology, there is no increase of sea-salt aerosol loading during dust intrusions, in contrast with the standard methodology. During the dust periods there is even a decrease in the contribution of sea salt to the aerosol that may result either from excessive calculation of soil $Na^+$ and $Mg^{2+}$ or, more probably, from an increased deposition rate of sea salt during dusty periods, by co-sedimentation of dust and sea-salt particles. Because of the application of our adapted alternative methodology, the amount of calculated sea-salt contribution decreases by 47 % in the dry-haze season and 32 % in the rest of the year.

The correct determination of sea-salt ion concentrations permits the estimation of the remaining common elements, supposedly from soil origin. In this way, it is possible to calculate $Mg_{soil}$, $K_{soil}$ and $Ca_{soil}$ concentrations. Determination of total $Na_{soil}$ and $Cl_{soil}$ is not feasible in this work because of analytical limitations.

Taking into account that we did not, or could not, measure with accuracy P and Na, the calculation of soil contribution was done by adapting Eq. (2) to

$$\text{Soil dust} = 1.15(2.14Si + 1.89Al + 1.43Fe + 1.39Mn \quad (55)$$
$$+ 1.67Ti + 1.66Mg_{soil} + 1.40Ca_{soil}$$
$$+ 1.20K_{soil}), \quad (3)$$

where the factor 1.15 is an average for the nine sites studied by Moreno et al. (2006) (see Table S1 in the Supplement for details and clarification). This factor is higher than the corresponding values for the average continental/upper crust (1.05–1.06), taken from Manson and Moore (1982) and Wedepohl (1995).

## 4.3 Secondary formation processes

The attribution of analysed water-soluble anions and cations to different sources and formation processes can be done using the sequential ionic balance proposed by Alastuey et al. (2005), adapted and developed by Mirante et al. (2014) for Madrid urban aerosol. The present situation, with very large inputs of dust and marine aerosols, imposes a further adaptation of the ionic balance method, because gas-to-particle reactions involving precursor pollutants and sea-salt spray, or dust, cannot be neglected, and from the evaluation of dust and sea-salt composition, the amounts of soluble ions of sea-salt and dust origin have to be initially calculated. Therefore, the ionic balance applied to the present samples is the following:

1. Start by calculating soil $Na_{soil}^+$ and soil $Mg_{soil}^{2+}$ from $Fe/Na^+$ and $Fe/Mg^{2+}$ edge lines in Fig. 2.

2. Calculate sea-salt $Na_s^+$ and $Mg_s^{2+}$ from differences between total and soil $Na_{soil}^+$ and $Mg_{soil}^{2+}$.

3. Recalculate sea-salt $Na_{ss}^+$ and $Mg_{ss}^{2+}$ using minimum values and the $Na^+/Mg^{2+}$ ratio in seawater.

4. Calculate sea-salt mass concentration and composition from $Na_{ss}^+$ and $Mg_{ss}^{2+}$, and seawater ion ratios, taking into account available $Cl^-$.

5. Calculate non-sea-salt $SO_4^{2-}$, $NO_3^-$ and $Cl^-$.

6. Associate, sequentially, free non-sea-salt $SO_4^{2-}$ and $NO_3^-$ with $NH_4^+$, until all $NH_4^+$ is neutralised.

7. From the differences between total and sea-salt cations, calculate soil $Na_{soil}^+$, $Mg_{soil}^{2+}$, $K_{soil}^+$ and $Ca_{soil}^{2+}$.

8. Associate free $NO_3^-$, sequentially, with free sea-salt and soil cations.

9. Associate, totally, $CO_3^{2-}$, sequentially, with free soil $Ca_{soil}^{2+}$, $Mg_{soil}^{2+}$, $Na_{soil}^+$ and $K_{soil}^+$.

10. Associate free $SO_4^{2-}$, sequentially, with free $Ca_{soil}^{2+}$, $Mg_{soil}^{2+}$, $Na_{soil}^+$ and $K_{soil}^+$.

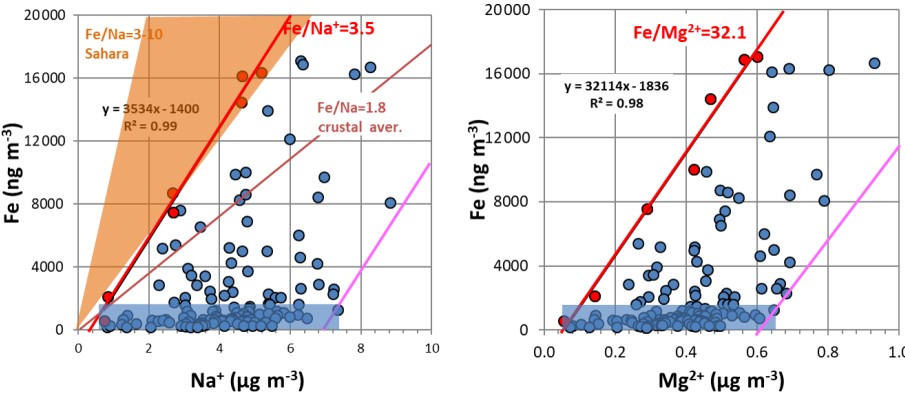

**Figure 2.** Edge lines (in red) for Fe versus $Na^+$ and Fe versus $Mg^{2+}$ intercomparisons. The blue rectangle represents periods without significant dust intrusions. The pink lines are parallels to the red edge lines in the maximum ion concentration regions. Also shown are Fe/Na ratio ranges in Sahara and global soils, for total sodium, taken from Moreno et al. (2006) and Manson and Moore (1982). Edge lines are best-fit linear lines traced through points in red.

11. Associate free $Cl^-$ with free $Na^+_{soil}$ and $Mg^{2+}_{soil}$. Any excess $Cl^-$ is associated with $K^+_{soil}$.

12. Calculate the total masses of water-soluble soil sulfate, nitrate and chloride.

13. Edge line ratios between Fe and sulfate, nitrate, or chloride permit a rough calculation of the fraction of these ionic compounds that is present in the soil or that results from secondary reaction with atmospheric produced acids (for visualisation, see Fig. S5 in the Supplement).

Using an Excel spreadsheet, the 13 steps were applied, sequentially, in order to attribute all measured cations and anions to sea salt, soil and secondary inorganic compounds. The first four steps were described in detail in the beginning of this section. Due to space limitations, only results for the remaining eight steps are presented.

Using a OM/OC ratio of 2.0 and measured EC and OC, it is also possible to estimate the total non-carbonated, carbonaceous aerosol.

With this IMB methodology, it is possible to account for the presence of seven source classes: sea-salt spray (SeaSalt), insoluble soil dust (SoilDust ins), soluble soil dust (SoilDust sol), secondary inorganic compounds from the reaction of atmospheric acids with ammonia (SIC am), sea salt (SIC ss) and dust (SIC du), and non-carbonate carbonaceous elemental and organic matter (carbon).

The results of water-soluble compounds are presented in Table 2 for the dry-haze and dust-free seasons. Concentrations of secondary ammonium salts (SIC am) are more than doubled during the dust-free season, probably as a result of higher temperatures and transport of air masses from non-desert polluted areas, or the removal by co-sedimentation with dust during dust episodes. Soluble soil dust (SoilDust sol) are more than tripled during the dry-haze season, being

formed mainly by calcium carbonates and sulfates, sodium nitrates and sulfates, and by sodium chloride.

Reaction of acid precursors with soil dust (SIC du) produces equivalent amounts of secondary compounds during the two seasons, probably because of limited availability of acidic precursors. Secondary processes resulting from acidic reactions with sea salt (SIC ss) produce more sea-salt secondary material during the dust-free season. As, in the present conditions, it is difficult to clearly differentiate between the two processes, because it is not possible to give a priority in the competitive acidic reaction with sea salt or dust; in the rest of the publication, the two source processes are treated together, as SIC ss plus du.

## 4.4 Particulate water

The fractional contribution of the six adapted source classes is given in Fig. 3 for the two seasons and for the total campaign. The figure reveals that the sum of the quantified sources only accounts for 76 % of the measured $PM_{10}$ total mass concentration, on average, during the total measurement campaign. The value decreases to 68 % during the dust-free season. These fractions are of the same order of values published in literature (Perrino et al., 2013; Rees et al., 2004; Andrews et al., 2000; Grigoratos et al., 2014).

As, in our case, carbonates were directly measured, it is predictable that most of the unaccounted mass results from the aerosol water content, in the form of adsorbed/absorbed water (hydrates in soil constituents were already considered with the application of factor $F$), now that $PM_{10}$ total mass measurement was performed at 20 °C and 50 % RH, in conditions of equilibrium between the laboratory atmospheric water vapour and the particulate material in the filter.

To estimate the amount of sorbed water in the aerosol, we consider that, by approximation, there is a thermodynamic equilibrium between the controlled room atmosphere,

**Table 2.** Soil and secondary inorganic compounds resulting from the ionic balance.

| Inorganic compounds | SIC am | | SoilDust sol | | SIC ss plus du | | | |
| --- | --- | --- | --- | --- | --- | --- | --- | --- |
| | | | | | SIC du | | SIC ss | |
| | Dust-free ($\mu g\,m^{-3}$) | Dry haze ($\mu g\,m^{-3}$) | Dust-free ($\mu g\,m^{-3}$) | Dry haze ($\mu g\,m^{-3}$) | Dust-free ($\mu g\,m^{-3}$) | Dry haze ($\mu g\,m^{-3}$) | Dust-free ($\mu g\,m^{-3}$) | Dry haze ($\mu g\,m^{-3}$) |
| $(NH_4)_2SO_4$ | 0.94 | 0.37 | | | | | | |
| $NH_4NO_3$ | 0.00 | 0.01 | | | | | | |
| $CaCO_3$ | | | 0.27 | 3.15 | | | | |
| $MgCO_3$ | | | 0.01 | 0.25 | | | | |
| $Na_2CO_3$ | | | 0.00 | 0.28 | | | | |
| $K_2CO_3$ | | | 0.00 | 0.00 | | | | |
| $NaNO_3$ | | | 086 [TS12] | 1.26 | 0.79 | 0.83 | 0.43 | 0.16 |
| $Mg(NO_3)_2$ | | | 010 [TS13] | 0.12 | 0.09 | 0.08 | 0.08 | 0.03 |
| $KNO_3$ | | | 0.06 | 0.10 | 0.05 | 0.06 | 0.01 | 0.00 |
| $Ca(NO_3)_2$ | | | 0.01 | 0.00 | 0.00 | 0.00 | 0.02 | 0.01 |
| $CaSO_4$ | | | 0.59 | 0.43 | 0.52 | 0.28 | | |
| $MgSO_4$ | | | 0.01 | 0.15 | 0.01 | 0.09 | | |
| $Na_2SO_4$ | | | 0.12 | 0.80 | 0.11 | 0.52 | | |
| $K_2SO_4$ | | | 0.05 | 0.06 | 0.05 | 0.04 | | |
| $NaCl$ | | | 0.27 | 1.75 | 0.13 | 0.0.29 [TS14] | | |
| $MgCl_2$ | | | 0.00 | 0.06 | 0.00 | 0.01 | | |
| $KCl$ | | | 0.00 | 0.06 | 0.00 | 0.01 | | |
| Total | 0.94 | 0.37 | 2.34 | 8.48 | 1.74 | 2.21 | 0.54 | 0.20 |

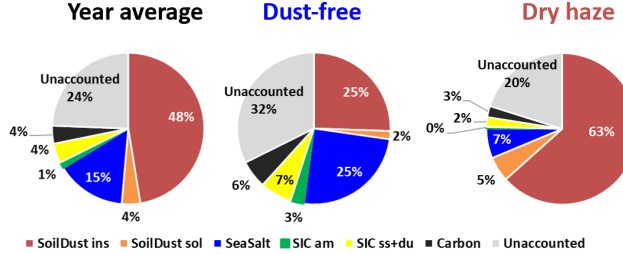

**Figure 3.** Contribution of different components to the $PM_{10}$ measured total mass, estimated by IMB, for the whole sampling campaign and for the dust-free and dry-haze seasons.

at 20 °C and 50 % RH, and the aerosol deposited on the filter, during the mass weighting in the laboratory, and also that the behaviour of different compounds is independent of internal or external mixture conditions.

For sea salt, thermodynamic information from Tang et al. (1997) was used, which, at 50 % RH and 20 °C, indicates a water/salt mass equilibrium ratio of 0.4 for a dry phase state, or 1.4 for a metastable deliquescent liquid phase state. The ISORROPIA thermodynamic equilibrium model (Nenes et al., 1998a, b) was applied, at 20 °C and 50 % RH, both to the calculated sea salt alone (SeaSalt) and considering together the sea salt and the secondary inorganic compound formation by the attack of atmospheric acids (SeaSalt plus SIC ss).

ISORROPIA output gave water fractions that were very similar, in both cases, and also coincident with the values taken from Tang et al. (1997) for wet or dry containing phases. As information about phase status during weighting is unavailable and the weighting was performed at RH very near the forced efflorescence point (47 % RH), we used for the water/salt ratio the arithmetic average of the two equilibrium values (0.9).

For secondary inorganic water-soluble ammonium salts (mainly ammonium sulfate), thermodynamic information from Xu et al. (1998) and Tang and Munkelvitz (1994) was applied, which shows an equilibrium water/ammonium sulfate mass ratio of 0.4 at 50 % RH.

The information concerning the water content of organic polar matter was taken from Speer et al. (2003) who used a water/OC ratio of 0.2.

Suspended soil also sorbs water, principally the water-soluble ionic component. We used ISORROPIA, version 2.1, which includes ions from crustal origin, to estimate the water content in soluble dust. The ISORROPIA II version was run for a liquid metastable assumption and applied to the soluble soil dust (SoilDust sol) and to the sum of soluble soil dust and secondary inorganic compounds formed from the attack of atmospheric acids on dust particles (SoilDust sol plus SIC du). In the first case, the average water/salt mass ratio calculated was 0.22 in the dust-free season and 0.47 in the

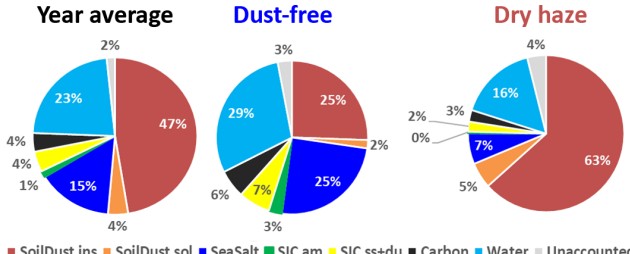

**Figure 4.** Inclusion of estimated water in the IMB for the total sampling campaign and for the dust-free and dry-haze seasons.

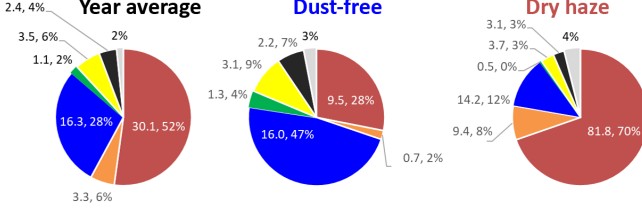

**Figure 5.** IMB obtained by attributing the calculated water content to the respective source classes for the entire campaign and the dust-free and dry-haze seasons. Values are in $\mu g\,m^{-3}$ and percentage of measured total mass.

dry-haze season. In the mixture, the calculated ratios were comparable (0.14 and 0.50, respectively). Due to the lack of more specific information, water/soluble dust ratio values of 0.2, during the dust-free season, and 0.5 during the rest of the year, were employed.

Various important components of the insoluble fraction of dust are hygroscopic, such as clay minerals. Schuttlefield et al. (2007) measured water adsorption by clay minerals having found a large variability in the water uptake by different clay mineral species, with water/clay mass ratios varying from 0.02 to 0.06 for kaolinite, going up to 0.17 for illite and 0.08–0.7 for montmorillonite, at 23 °C and 50 % RH. Based on these data, a round value of 0.1 for the ratio of water/SoilDust ins was adopted for our samples.

The estimation of total water content in the collected aerosols using the above referred ratio assumptions is presented in Fig. 4. The figure shows that, by using this methodology, there is almost a perfect account of the $PM_{10}$ total mass. Water represents an average contribution of 23 % to the aerosol mass. During the dry-haze period, the calculated water, on average, accounts for 16 % of the $PM_{10}$ mass, with a maximum contribution of 29 % during the dust-free season. By including particulate water, only around 2 %–4 % of the PM total mass is unaccounted, with the applied IMB methodology.

### 4.5 Comparison with PMF

The final ionic and mass balance calculations are presented in Fig. 5 for the total campaign and the two seasons, considering the components associated with the respective water uptake. The addition of sorbed water reinforces the impact of hygroscopic/soluble components, such as sea salt, in the atmospheric aerosol loading, which, for example, during the dust-free season increases its contribution from 25 % to 47 %.

The results of IMB can be evaluated and compared with PMF results applied to the same data set and already published by Salvador et al. (2016). Here, the published PMF results were reorganised, in order to make explicit the PMF contributions during the two dry-haze and dust-free seasons

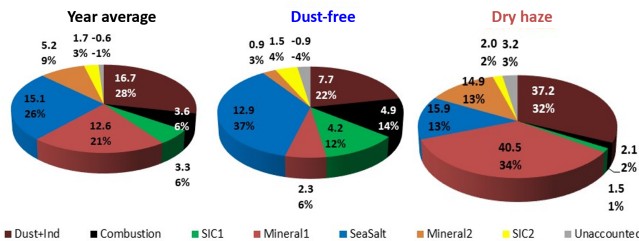

**Figure 6.** PMF source apportionment results for the total campaign and during the two pollution seasons. Values are in $\mu g\,m^{-3}$ and percentage of measured total mass.

and to show unaccounted/excess calculated PM mass, as represented in Fig. 6.

The PMF analysis could differentiate seven aerosol sources: three concerning soil contamination and two considering secondary inorganic components, plus sea-salt and combustion processes. The dust sources comprised "Mineral1", associated mainly with Al, Si and Fe; "Mineral2", associated with $CO_3^{2-}$ and $Ca^{2+}$; and "Dust plus Ind", containing both soil elements (Al, Si, Fe) and tracers of industrial emissions (V, Ni, Cu and Cr). The secondary inorganic components included "SIC1", associated with $NH_4^+$, $SO_4^{2-}$, $NO_3^-$ $Na^+$, $K^+$ and $Mg^{2+}$; and "SIC2", containing $SO_4^{2-}$, $NO_3^-$, $Ca^{2+}$ and $Mg^{2+}$. The "SeaSalt" source represented most of the variability of $Na^+$, $Cl^-$, $Mg^{2+}$, Br and $K^+$. The "combustion" source included mainly EC and OC variability (complementary information concerning PMF sources compositions and contributions can be found in Fig. S8 in the Supplement).

Figures 5–6 and Table 3 compare both methodologies for the two seasons and the total campaign (for individual sampling events, consult Figs. S6 and S7 in the Supplement). There is a good agreement between the two source apportionment techniques. Both methodologies reproduce almost totally the measured $PM_{10}$ total mass.

The figures and table show a good comparability between total soil dust estimated by both methods in any season. In the IMB, the SoilDust sol fraction is approximately equivalent to the Mineral2 component of PMF. The mass balance

**Table 3.** Comparison between IMB and PMF results, grouped into four main classes. PM$_{10}$ represents gravimetric measurements of total mass (in IMB: soil is the total of WetSoilDust ins plus WetSoilDust sol, SIC is the total of WetSIC am plus WetSIC ss plus du; in PMF: SIC is the total of SIC1 plus SIC2, soil is the total of Mineral1 plus Mineral2 plus Dust plus Ind).

|  | Year average | | Dry haze | | Dust-free | |
|---|---|---|---|---|---|---|
|  | PMF ($\mu$g m$^{-3}$) | IMB ($\mu$g m$^{-3}$) | PMF ($\mu$g m$^{-3}$) | IMB ($\mu$g m$^{-3}$) | PMF ($\mu$g m$^{-3}$) | IMB ($\mu$g m$^{-3}$) |
| Soil | 34.5 | 33.5 | 92.6 | 91.2 | 10.9 | 10.2 |
| Sea salt | 15.1 | 16.3 | 15.9 | 14.2 | 12.9 | 16.0 |
| SIC | 4.9 | 4.6 | 3.5 | 4.2 | 5.7 | 4.4 |
| Carbon/combust | 3.6 | 2.4 | 2.1 | 3.1 | 4.9 | 2.2 |
| Sum | 58.2 | 56.8 | 114.1 | 112.7 | 34.5 | 32.8 |
| PM$_{10}$ | 57.7 | | 117.3 | | 33.9 | |

method could not discriminate the Dust plus Ind from the total insoluble dust fraction, as was possible with PMF.

Contribution of sea salt was also equally estimated by the two techniques, on average, during the dry-haze season, but during the dust-free period the IMB estimated somehow higher sea-salt levels than the PMF.

SIC values are similar in both source apportionment methodologies, but during the dust-free period PMF estimated higher SIC contributions. This was mainly due to higher estimations of ammonium secondary salts by PMF, which only can happen if other compounds are included in the source component, as evidenced by the PMF source profile.

There is also a higher contribution from carbonaceous/organic/combustion material in PMF, by comparison with IMB, during the dry season, although a high factor of 2 was applied to the OM/OC ratio in the mass balance approach. The inclusion by the PMF of other constituents from combustion processes in west Africa is, probably, the reason for the discrepancy.

Further insight into the capabilities and limitations of IMB versus PMF can be attained by comparing source classes for each individual sample, as presented in Fig. 7. From this figure it is possible to confirm the good comparability between the soil source estimations by both methods, with a linear ratio estimation of 1.04 and a correlation $R = 0.97$.

For sea-salt estimation, the comparison is not so good with IMB/PMF ratio estimation of only 0.68 and an $R = 0.82$. This happens principally because, for several samples, PMF gives zero or negative sea-salt contributions, while the IMB estimates important sea-salt values. For a location in the middle of the ocean, it is not expectable to have absence of sea salt, and therefore, in our opinion, PMF fails, attributing, probably, the available sea salt to a soil source. At high concentration levels, there is a tendency of PMF to give higher sea-salt values than IMB. An inspection of PMF compounds' contribution to the sea-salt source shows that, on average, there is a mass inclusion of about 20 % of elements such as Si, Al, Fe, etc. in this source (see Fig. S8, graph 8, for clarification). Therefore, it is clear that PMF could not completely separate soil from sea-salt sources, probably because of the overwhelming presence of soil during dust episodes. During the dust-free season, IMB tends to give somehow higher sea-salt contributions than PMF (sea-salt IMB is 0.73 sea-salt PMF plus 6.8; $R = 0.84$). One of the possible reasons may be a too-high estimation of sea-salt sorbed water in IMB.

Both methodologies also compare reasonably well in what concerns secondary inorganic compounds (SICs) contributions to the aerosol loading, with a IMB/PMF ratio of 0.83 and a correlation coefficient $R = 0.77$. Where the comparison fails completely is in the carbonaceous (IMB) versus combustion (PMF) sources that present no clear relation. This results from several facts as exposed in the following text. The IMB source profile represents only non-carbonate carbonaceous matter, irrespective of the source, while the PMF factor intends to represent all emissions from combustion sources, besides carbonaceous matter. Therefore, from Fig. 7, it is possible to observe in several samples important contributions of carbonaceous matter estimated by IMB, while PMF gives zero to negative combustion emission estimations. Most probably this results from soil contribution to organic matter that in PMF is attributed to dust or anthropogenic sources (Ant plus Dust; see graphs C and H in Fig. A8 CE1 for clarification). Also, in PMF, the combustion source has, on average, an important contribution (around 50 %; see Fig. A8 CE2 for clarification) of elements, such as Si, Al, Fe, etc., from soil origin; therefore, in our opinion, this PMF combustion source is highly contaminated with soil, with PMF not fully capable of separating combustion from soil dust, due to the overwhelming presence of soil particles during dust episodes.

From Table 3 and Fig. 7 we may then conclude that the IMB solution compares well with PMF for dust and SIC, but the two methods show important discrepancies for individual samples, principally in the estimation of sea salt and carbonaceous aerosol contributions.

**Atmos. Chem. Phys., 18, 1–16, 2018**                                                     **www.atmos-chem-phys.net/18/1/2018/**

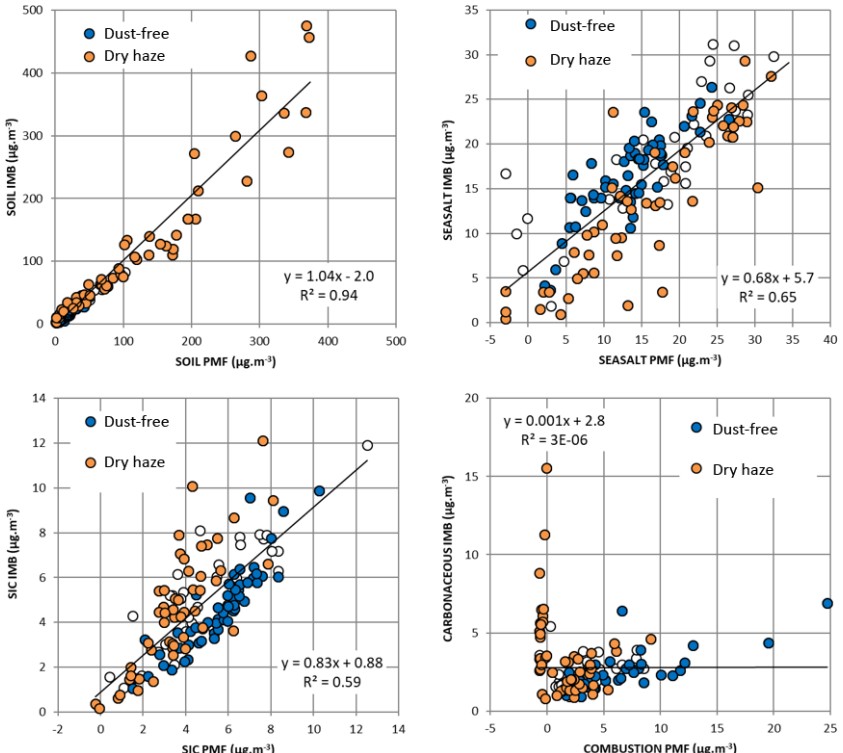

**Figure 7.** Comparison between the apportionment of individual samples using IMB and PMF for four source classes. Circles filled in white represent periods with intermittent dust intrusions. Linear best fits are presented for the total sampling campaign.

## 4.6 Accuracy and errors

IMB and PMF are subject to a number of errors that affect the precision and accuracy of the sources' estimation. Fully accounting for all errors is difficult, because some used information (bibliographic or experimental) has no available accuracy or is subject to unknown and unexpected errors. PMF applies several statistical tests to evaluate the influence of random errors and the rotational ambiguity of the obtained solution. Although these tests indicated that the PMF solution was robust, they could not identify collinearity problems that resulted in the significant contamination of combustion and sea-salt sources with soil dust.

Both methodologies are equally affected by errors in the aerosol chemical analysis. Probably the higher relative analytical errors are related with elements and EC evaluation, in conditions of high dust concentrations, although care was taken to sample for shorter periods during dust episodes. EC is frequently near the detection limits and it is quite difficult to fully evaluate the interference of coloured dust during the thermo-optical analytical process. This affects, in an unknown value, principally, the evaluation of the PMF combustion source that uses EC as its principal tracer (see Fig. S8 for clarification).

In IMB, there are probably four estimations where the errors influencing the source apportionment are higher: (a) cal-

culation of water sorbed in sea salt; (b) the estimation of total soil content based in factor $F$; (c) calculation of sea-salt $Na^+$; (d) the estimation of organic matter from OC. It is necessary to take into account, however, that total errors are controlled by measured $PM_{10}$ total mass constraint (Malm et al., 1994 use this comparison as a validation and self-consistency check for their reconstructed aerosol mass balance). Falling CE3 the sum of estimated masses within 96 %–98 % of measured $PM_{10}$ total mass, the individual errors are probably limited (or of opposite directions, with mutual compensation).

Estimation of water content in sea salt depends mainly on metastable equilibrium considerations that give a water/dry sea-salt mass ratio varying from 0.4 to 1.4. An average value of 0.9 was applied in our calculations. Application of the two extreme values would vary the fractional contribution of wet sea salt by approximately ±6 %, ±7 % or ±3 % for the total sampling campaign, the dust-free or the dry-haze periods. It is necessary to consider, however, that by choosing the two ratio extremes the closure of the total PM mass would be affected, resulting in a maximum unaccounted mass of 16 % for the choice of a 0.4 value, and a maximum overprediction of 8 % for the choice of 1.4 ratio, and therefore the correctness of these extreme values is questionable.

The calculation of the total soil dust was based on Eq. (3) that uses an average factor $F$ value of 1.15. $F$ values taken

from Moreno et al. (2006) (see Table S1 in the Supplement, for clarification) vary in the range 1.09–1.27 (standard deviation of 0.21) and the factor $F$ for global composition is 1.05–1.06. A range of approximately 16 % will result from the calculation of soil dust contribution by application of the two extreme $F$ factors. Use of extreme $F$ values would give a maximum unaccounted PM mass of 8 % or an excess total PM mass calculation of 6 %.

The Fe/Na$^+$ edge line methodology used to estimate Na$^+$ and Mg$^{2+}$ was conservative, minimising the subtraction from sea salt. This was partially compensated by using minimum values and the Na$^+_{ss}$/Mg$^{2+}_{ss}$ ratio in seawater. The edge line intercepts the $x$ axis at Na$^+$ levels within the range of values observed for periods without significant dust intrusions, in accordance with the fact that there is always sea-salt spray in this marine atmosphere (see Fig. 2a, where air masses without dust intrusions are represented by a blue rectangle). The points to the right of the edge line have excess Na$^+$, by comparison with Fe in the edge line. This excess can have two origins: either it results from variability in the relative content of Na$^+$ in soil dust, or it is resultant from the variability in the contributions of sea salt. It is clear that the points at low Fe levels, within the blue rectangle, are only consequence from variability in sea-salt spray loading. On the right border of the blue rectangle, a pink straight line is drawn, parallel to the Fe/Ion edge lines. As all the measured points are within both lines, this is a strong indicator that Na$^+$ and Mg$^{2+}$ increase, in relation to the edge lines, result mostly from variability in sea-salt input. The real value of the errors in this methodology is difficult to establish, but a change in the Fe/Na$^+$ edge ratio of 10 % would have no visible effect in the estimation of sea salt or soil dust by IMB.

The estimated value for non-carbonate carbonaceous matter depends strongly from the OM/OC ratio. In the literature, values in the range 1.2–2.3 have been proposed. We used a ratio of 2.0 in the high end of the range because the sampling was performed at a background location, away from primary combustion sources, with plenty of time for oxidation and secondary formation. However, during dust episodes, important fractions of the organic material have a soil origin and for that less reliable information exists. As the carbonaceous fraction in PM mass is only 2 %–6 %, errors in OM/OC ratio have only a small effect in total mass attribution, but important errors, of the order of 40 %, can result in the estimation of the carbonaceous mass if a OM/OC ratio of 1.6 is the correct assumption, instead of 2.0.

## 5   Conclusions

Atmospheric aerosol was collected during 1 year, as PM$_{10}$, in air masses transported from north Africa to Cabo Verde islands and submitted to total mass, elemental, ionic and carbon content analysis. Two clear different aerosol seasons were observed: one in December–February, with frequent intrusions of dust from Africa (denoted dry haze), and the other in May–September, without direct African dust contamination (dust-free).

The application of IMB to the collected aerosol permitted the determination of six to seven source classes: insoluble dust, soluble dust, sea salt, secondary inorganic compounds from the reaction of atmospheric acidic precursors with sea salt, dust and ammonia, and carbonaceous matter.

The sum of calculated components only partially closed the mass balance, being 20 %–30 % of the measured PM$_{10}$ total mass unaccounted. Consideration and estimation of particulate water content, based in bibliographic and thermodynamic assumptions, permitted an almost total closure of the mass balance. This outcome is diverse from most previous mass balance studies, such as those referred to in the section on mass balance methodologies, where mass closure was only partial. Therefore, the present IMB methodology permits a more well-based mass account and apportionment of formation processes and sources.

During the dry-haze season, dust contributed with around 80 % to the aerosol mass loading, while in the dust-free period the main aerosol component was sea salt that constituted approximately 50 % of the aerosol mass.

The IMB methodology was compared with PMF results applied to the same data set. In seasonal averaged terms, the outcomes of the two methodologies were comparable for the most important sources and formation processes. Comparison between individual samples showed, however, significant differences, principally for the sea-salt spray and the carbonaceous/combustion sources. Because of the overwhelming presence of dust in most samples, the PMF could not clearly separate dust from sea-salt sources. On the other hand, IMB could not discriminate soil organic matter from combustion emissions.

We can rely on the complementarity of both methods for the evaluation of sources contributing to atmospheric contamination, in circumstances of very high natural inputs of sea salt and desert dust particles, subject to atmospheric transformation during long-range transport. Utilisation of these two independent source apportionment methodologies adds confidence to the apportionment of an atmospheric aerosol with quite specific and uncommon characteristics.

Otherwise, source composition and contribution knowledge obtained with IMB can be used to complement the constrains already applied in the last versions of the PMF model (Amato and Hopke, 2012; Liu et al., 2015; Chen et al., 2018), in order to get more refined solutions than the original ones. Constraints can be created using specific ratios between two different species or mass balance equations derived from IMB techniques such as those performed in this study.

*Data availability.*   TS15

**The Supplement related to this article is available online at https://doi.org/10.5194/acp-18-1-2018-supplement.**

*Author contributions.* .TS16

*Competing interests.* The authors declare that they have no conflict of interest.TS17

*Acknowledgements.* The authors gratefully acknowledge the Portuguese Science Foundation through the project CV-DUST – Atmospheric aerosol in the Cape Verde region: carbon and soluble fractions of PM$_{10}$ (PTDD/AAC-CLI/100331/2008) and the PhD scholarship of João Cardoso. C2TN/IST/ULisboa authors gratefully acknowledge the FCT support through the UID/Multi/04349/2013 project.

Edited by: Chak K. Chan
Reviewed by: three anonymous referees

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

**Remarks from the language copy-editor**

CE1     Please check. There is no Fig. A8 in this paper.
CE2     Please check. There is no Fig. A8 in this paper.
CE3     Please note that "falling" is not grammatical in the given context. Please revise this term.

**Remarks from the typesetter**

TS1     Please provide department.
TS2     The composition of all figures has been adjusted to our standards.
TS3     Copernicus Publications collects the DOIs of data sets, videos, samples, model code, and other supplementary/underlying material or resources as well as additional outputs. These assets should be added to the reference list (author(s), title, DOI, and year) and properly cited in the article. If no DOI can be registered, assets can be linked through persistent URLs. This is not seen as best practice and the persistence of the URL must be secured.
TS4     Not mentioned in the reference list.
TS5     Please provide the short title.
TS6     Not mentioned in the reference list.
TS7     Not mentioned in the reference list.
TS8     Please confirm change to bold for visibility.
TS9     Please confirm.
TS10     Should this be 2014? Otherwise, it is not mentioned in the reference list.
TS11     Not mentioned in the reference list.
TS12     Should this be 0.86?
TS13     Should this be 0.10?
TS14     Should this be 0.29?
TS15     Please provide a statement on how your underlying research data can be accessed. If the data are not publicly accessible, a detailed explanation of why this is the case is required. The best way to provide access to data is by depositing them (as well as related metadata) in reliable public data repositories, assigning digital object identifiers (DOIs), and properly citing data sets as individual contributions. Please indicate if different data sets are deposited in different repositories or if data from a third party were used. If no DOI is available, assets can be linked through persistent URLs to the data set itself (not to the repositories' home page). This is not seen as best practice and the persistence of the URL must be secured.
TS16     Please note that the section "Author contributions" is mandatory for ACP.
TS17     Declaration of all potential conflicts of interest is required by us as this is an integral aspect of a transparent record of scientific work. If there are possible conflicts of interest, please state what competing interests are relevant to your work.
TS18     Please confirm.
TS19     Please provide last access date.
TS20     Please provide page range or article number.
TS21     Not mentioned in the text.
TS22     Not mentioned in the text.
TS23     Please confirm.
TS24     Please confirm or provide different location.
TS25     Please provide publisher location.