# Peer review of "Source apportionment of atmospheric aerosol in a marine dusty environment by Ionic/composition Mass Balance (IMB)"

_Atmospheric Chemistry and Physics, 2018_

## Referee Comment (RC1) · Anonymous Referee #2 · 19 Mar 2018

Some improvements have been made in this revised version and I recommend publication in ACP. The paper reported the development of mass balance models for source apportionment of aerosol particles and the models were compared with PMF model. Results showed good agreements between the two models. I have following questions about the ionic mass balance (IMB) model as highlighted below.

(1) Is that possible to compare the mass balanced model developed in the present study with previous works, such as Malm et al., 1994 (JGR, 99, 1347), and other mass balance model thereby to further demonstrate the usefulness of new model. (2) I would suggest to add some discussions about the robustness of the new IMB model

[Figure]

as compared with PMF. (3) Authors mentioned uncertainties in multivariate methods, such as PMF, are there any uncertainties in IMB model developed and applied in this study?

---

## Referee Comment (RC2) · Anonymous Referee #3 · 28 Mar 2018

This paper analyzed elemental and some ion data in PM10 acquired from Cape Verde, an island west of the African Continent, with the attempt to determine the contributions from Saharan dust, sea salt, and their derivatives due to atmospheric transformation. Their approach is mainly based on ion balance, assuming known crustal and sea water chemical composition and the sequence of cation/anion neutralization. The mass and ion closure result from the approach generally make sense, especially when accounting for the residual water content. However, uncertainties were not provided for each contribution estimate, as PMF and other receptor models usually did. In many cases, the authors picked a middle value from a range of possible ratios, such as the water/soluble dust ratio, $Fe/Na^+$ ratio, $Mg^{2+}ss/Na^+ss$, ratio, etc. to carry out their cal-

culation. Is it possible to propagate uncertainties in these assumptions throughout the calculation and give an overall uncertainty estimate in Table 3? The uncertainties should be compared with those from PMF based on the bootstrapping or DISP methods. In fact, the dust Fe/Na+ ratio (3.7) determined from the edge line (Figure 2) is based on only 3 points and the ratio falls at the low end of possible range of Fe/Na ratio for Sahara dust. It is nearly impossible to find a period with zero sea salt contribution at the island, and therefore using the edge line to determine the dust Na+ fraction is risky. At the very least, uncertainty in this method should be given and propagated into all subsequent calculations.

When the authors compare IMB and PMF source contribution, they only compare the average contributions for the three periods. It is also meaningful to compare: 1) correlation of respective source contributions determined by the two methods across individual samples and 2) chemical composition of corresponding sources/classes, particularly the sea salt and dust sources, to examine whether the authors' assumptions in IMB, such as the aforementioned Fe/Na+ ratio, are consistent with PMF. These comparisons should be presented and discussed explicitly.

Additional comments:

Page 10, Line 16. What is the range of this F factor? Can this be propagated into the uncertainty estimate?

Page 10, Line 33. Does K+ (and perhaps Cl-) generated from biomass burning need to be considered in the IMB for the region?

Page 13, Line 12-19. How does the assumption of average inorganic and organic water growth factor influence the mass closure, and the contribution of each source to PM10? Can the range be estimated?

---

## Referee Comment (RC3) · Anonymous Referee #1 · 2 Apr 2018

This study applied a method called IMB for source apportionment in Cape Verde, and compared the results with those from PMF. Essentially, the IMB method is nothing more than linear combinations of weighted concentrations of PM components. It might be acceptable to identify some factors with known and relatively fixed ratios between species, e.g. Na and Cl in sea spray and metals and oxygen in dust, however, it is of limited use when handling sources with similar components but varied weights. The factor of "non-carbonate Carbonaceous elemental and organic matter" showed exactly the limitation of this IMB method, as OC and EC may come from multiple anthropogenic sources but this method failed to separate them. OC is additionally contributed by secondary process which was also not able to identify by IMB. If performing well, PMF should be able

to identify the multiple sources for OC and EC, as evidenced in many other places in the world. The fact that the IBM results happen to be consistent with PMF ones is that the PM contributing sources in Cape Verde is quite simple, which cannot sufficiently justify that this method is also applicable in other places with complicated contributing sources. In contrast, PMF has seen successful applications around the world with distinct environmental conditions. Therefore, I consider the approach applied in this study had significant drawback, and the limited scientific significance of this study does not meet the standard of Atmospheric Chemistry and Physics.

---

## Author Comment (AC1) · 30 Apr 2018

These authors answers, together with changes added to the initial manuscript are also present in the annexed pdf file.

Anonymous Referee #1

This study applied a method called IMB for source apportionment in Cape Verde, and compared the results with those from PMF. Essentially, the IMB method is nothing more

than linear combinations of weighted concentrations of PM components. It might be acceptable to identify some factors with known and relatively fixed ratios between species, e.g. Na and Cl in sea spray and metals and oxygen in dust, however, it is of limited use when handling sources with similar components but varied weights.

Answer: We do not agree with the referee. Of course that the IMB is not a full solution to Source Apportion the atmospheric aerosol. However, partial mass balance has been used in the past in a multitude of situations to partially apportion the atmospheric aerosol with success. Here we extend this methodology further in order to apportion the totality of the aerosol mass, obtaining aerosol fractions that fit completely in the total PM mass and that give interesting and important information concerning sources and formation processes. This information is complementary to information obtained from PMF and is quite useful to test the accuracy of the PMF obtained solution. The "European Guide on Air Pollution Source Apportionment with Receptor Models" (http://publications.jrc.ec.europa.eu/repository/bitstream/JRC83309/lb-na-26-080-en-n.pdf ) clearly alleged in page 19 that "Receptor Models can be used in combination with independent methodologies (e.g. emission inventories, chemical transport models (CTMs)) to achieve more robust estimations by mutual validation of the outputs". In this paper we wanted to show that IMB may help in the obtaining of a more real and accurate solution than that the one obtained only with PMF. This will be especially important when the various source emission impacts in the receptor covary in time, situation in which methodologies such as PMF are unable of completely separating the different sources of Secondary Inorganic Aerosol (SIA), as it was the present case. We would like to emphasize that in the review of ambient particulate matter source apportionment results obtained using receptor models achieved by Belis et al. (Atmospheric Environment 69 (2013), 94-108) it was stressed that "SIA" was the strongest source for PM mass concentrations over Europe. These authors also declared that "SIA" contributions increased significantly when PM10 and PM2.5 concentrations, respectively were above the normative limit values. However, in most of the studies that were reviewed "SIA" was mainly composed of ammonium- sulphate and nitrate

and only one third of them reported sulphate and nitrate contributions separately. It should also be noted that SIA levels thus determined could be underestimated since other secondary inorganic contributions were included in the classical "Sea/Road Salt" and the "Crustal/Mineral Dust" categories (Belis et al., Atmospheric Environment 69 (2013), 94-108; Atmospheric Environment 85 (2014), 275-276). From these results it can be concluded that in general terms RMs currently used were not able to provide a rather-full discrimination of different secondary inorganic contributions to the PM mass registered at a given receptor site.

The factor of "noncarbonate Carbonaceous elemental and organic matter" showed exactly the limitation of this IMB method, as OC and EC may come from multiple anthropogenic sources but this method failed to separate them. OC is additionally contributed by secondary process which was also not able to identify by IMB. If performing well, PMF should be able C1to identify the multiple sources for OC and EC, as evidenced in many other places in the world.

Answer: We agree with the referee in relation to the limitations of the IMB method to completely separate and identify all the sources responsible for the carbonaceous matter, because these sources contain other substances besides carbonaceous material. But we continue to disagree with the referee about the capability of the IMB method to provide useful information concerning carbonaceous sources and formation processes. In the present case we did not invest much in separating the non-carbonated, carbonaceous component into fractions, because concentrations are quite low and, for EC, very near the limit of detection of the system analysis capabilities (very low concentrations of EC in a matrix with quite high concentrations of interfering colored dust). In the present case, the total contribution of non-carbonate carbonaceous matter to PM10 is only of the order of 2%. However in another situation where we are applying IMB (concerning urban pollution), where contribution of carbonaceous mater is of the order of 50%, we could separate the carbonaceous matter into biomass burning, primary fossil fuel emission by cars and secondary formation processes. For that, we

used levoglucosan measurements, edge lines for EC, versus OC versus fine K, versus levoglucosan and known ratios taken from specialized literature. In this urban data the biomass burning mass contribution compared very well between IMB and PMF (results to be submitted for publication, soon). This is a demonstration of the capabilities of IMB in "more complex" situations.

The fact that the IBM results happen to be consistent with PMF ones is that the PM contributing sources in Cape Verde is quite simple, which cannot sufïñĄciently justify that this method is also applicable in other places with complicated contributing sources. In contrast, PMF has seen successful applications around the world with distinct environmental conditions. Therefore, I consider the approach applied in this study had signiïñĄcant drawback, and the limited scientiïñĄc signiïñĄcance of this study does not meet the standard of Atmospheric Chemistry and Physics.

Answer: Part of the answer to this commentary has been given in the two previous answers. Here we will only complement the previous exposition. First, we want to stress that we do not have the intention of showing that the IMB should substitute the PMF, neither that it is better. We defend that IMB is a good methodology that extends and deepens frequently used methodologies in the past, which by being able of explaining the total PM mass is therefore more constrained and secured than previous incomplete attempts and that is complementary and useful to evaluate PM sources and formation processes in conjunction with other employed methods, such as PMF. The present evaluated aerosol is simple in what concerns the number of source contributions, taking into account classic air pollution considerations. But it is complex in which concerns collinearity of concentration variations because anthropogenic industrial and transport pollution are originated from the same directions than the sources of dust. Also, dust is emitted with different compositions in different parts of Africa and is important to know the dust contributions from each region. For telling the truth, neither the IMB, nor the PMF, were capable of fully discriminating the regional origins and contributions of the African dust. As demonstrated in the text now added to the manuscript, in the present

situation the PMF could not completely separate even sources such as Sea-salt and Dust, because of the common composition of dust and sea salt, concerning major ions. In many occasions PMF calculated zero sea salt contribution (impossible for a place in the middle of the ocean); in other occasions PMF considered important contributions of Al and Si in the sea salt component. So, this is not a "quite simple" situation.

Anonymous Referee #2

Some improvements have been made in this revised version and I recommend publication in ACP. The paper reported the development of mass balance models for source apportionment of aerosol particles and the models were compared with PMF model. Results showed good agreements between the two models. I have following questions about the ionic mass balance (IMB) model as highlighted below. (1) Is that possible to compare the mass balanced model developed in the present study with previous works, such as Malm et al., 1994 (JGR, 99, 1347), and other mass balance model thereby to further demonstrate the usefulness of new model.

Answer: The reference in consideration was added to the paper and several sentences were added to the text comparing IMB with previously published Mass Balance attempts.

(2) I would suggest to add some discussions about the robustness of the new IMB model C1as compared with PMF.

Answer: Text is added to the manuscript where the subject is discussed.

(3) Authors mentioned uncertainties in multivariate methods, such as PMF, are there any uncertainties in IMB model developed and applied in this study?

Answer: We have added a subsection in the manuscript where uncertainties for the IMB method are presented and discussed.

Anonymous Referee #3

This paper analyzed elemental and some ion data in PM10 acquired from Cape Verde, an island west of the African Continent, with the attempt to determine the contributions from Saharan dust, sea salt, and their derivatives due to atmospheric transformation. Their approach is mainly based on ion balance, assuming known crustal and sea water chemical composition and the sequence of cation/anion neutralization. The mass and ion closure result from the approach generally make sense, especially when accounting for the residual water content. However, uncertainties were not provided for each contribution estimate, as PMF and other receptor models usually did. In many cases, the authors picked a middle value from a range of possible ratios, such as the water/soluble dust ratio, Fe/Na+ ratio, Mg2+ss/Na+ss, ratio, etc. to carry out their calculation. Is it possible to propagate uncertainties in these assumptions throughout the calculation and give an overall uncertainty estimate in Table 3? The uncertainties should be compared with those from PMF based on the bootstrapping or DISP methods. Answer: We have added a subsection in the manuscript where uncertainties for the IMB method are presented and discussed.

In fact, the dust Fe/Na+ ratio (3.7) determined from the edge line (Figure 2) is based on only 3 points and the ratio falls at the low end of possible range of Fe/Na ratio for Sahara dust. It is nearly impossible to find a period with zero sea salt contribution at the island, and therefore using the edge line to determine the dust Na+ fraction is risky. At the very least, uncertainty in this method should be given and propagated into all subsequent calculations.

Answer- In this point we do not agree with the referee. This methodology is much better than the known alternative, which is to attribute all Na+ to sea salt production. In reality the edge line is not based in 3 points only (see Figure presented in the annexed pdf file). If we use an estimation taking into account predicable measuring analytical errors, the Fe/Na+ edge ratio is based in 10 points. As important, the edge line intercepts the x axe at Na+ levels within the range of values (represented by the blue rectangle) referring to periods without significant dust intrusions, in accordance with the fact that there is

always sea salt spray in this marine atmosphere (The subject is further discussed in the new added sub-section 4.5).

When the authors compare IMB and PMF source contribution, they only compare the average contributions for the three periods. It is also meaningful to compare: 1) correlation of respective source contributions determined by the two methods across individual samples and 2) chemical composition of corresponding sources/classes, particularly the sea salt and dust sources, to examine whether the authors' assumptions in IMB, such as the aforementioned Fe/Na+ ratio, are consistent with PMF. These comparisons should be presented and discussed explicitly.

Answer: New figures with individual comparisons between IMB and PMF (Figure 6a, b, c and d): Comparisons were discussed in added text, showing the capabilities of IMB to source apportion the aerosol.

Additional comments:

Page 10, Line 16. What is the range of this F factor? Can this be propagated into the uncertainty estimate?

Answer: The subject is discussed in new sub-section 4.5 (see annexed file).

Page 10, Line 33. Does K+ (and perhaps Cl-) generated from biomass burning need to be considered in the IMB for the region?

Answer: The amounts of possible bio-mass burning material are so small by comparison with other sources that any tentative to detail this source is mostly speculative.

Page 13, Line 12-19. How does the assumption of average inorganic and organic water growth factor influence the mass closure, and the contribution of each source to PM10? Can the range be estimated?

Answer: The subject is discussed in new added 4.5 sub-section (see annexed file).

Please also note the supplement to this comment:
https://www.atmos-chem-phys-discuss.net/acp-2018-10/acp-2018-10-AC1-supplement.pdf

**Supplement:**

This study applied a method called IMB for source apportionment in Cape Verde, and compared the results with those from PMF. Essentially, the IMB method is nothing more than linear combinations of weighted concentrations of PM components. It might be acceptable to identify some factors with known and relatively fixed ratios between species, e.g. Na and Cl in sea spray and metals and oxygen in dust, however, it is of limited use when handling sources with similar components but varied weights.

Answer: We do not agree with the referee. Of course that the IMB is not a full solution to Source Apportion the atmospheric aerosol. However, partial mass balance has been used in the past in a multitude of situations to partially apportion the atmospheric aerosol with success. Here we extend this methodology further in order to apportion the totality of the aerosol mass, obtaining aerosol fractions that fit completely in the total PM mass and that give interesting and important information concerning sources and formation processes. This information is complementary to information obtained from PMF and is quite useful to test the accuracy of the PMF obtained solution. The "European Guide on Air Pollution Source Apportionment with Receptor Models" (http://publications.jrc.ec.europa.eu/repository/bitstream/JRC83309/lb-na-26-080-en-n.pdf ) clearly alleged in page 19 that "Receptor Models can be used in combination with independent methodologies (e.g. emission inventories, chemical transport models (CTMs)) to achieve more robust estimations by mutual validation of the outputs". In this paper we wanted to show that IMB may help in the obtaining of a more real and accurate solution than that the one obtained only with PMF. This will be especially important when the various source emission impacts in the receptor covary in time, situation in which methodologies such as PMF are unable of completely separating the different sources of Secondary Inorganic Aerosol (SIA), as it was the present case. We would like to emphasize that in the review of ambient particulate matter source apportionment results obtained using receptor models achieved by Belis et al. (Atmospheric Environment 69 (2013), 94-108) it was stressed that "SIA" was the strongest source for PM mass concentrations over Europe. These authors also declared that "SIA" contributions increased significantly when PM10 and PM2.5 concentrations, respectively were above the normative limit values. However, in most of the studies that were reviewed "SIA" was mainly composed of ammonium- sulphate and nitrate and only one third of them reported sulphate and nitrate contributions separately. It should also be noted that SIA levels thus determined could be underestimated since other secondary inorganic contributions were included in the classical "Sea/Road Salt" and the "Crustal/Mineral Dust" categories (Belis et al., Atmospheric Environment 69 (2013), 94-108; Atmospheric Environment 85 (2014), 275-276). From these results it can be concluded that in general terms RMs currently used were not able to provide a rather-full discrimination of different secondary inorganic contributions to the PM mass registered at a given receptor site.

The factor of "noncarbonate Carbonaceous elemental and organic matter" showed exactly the limitation of this IMB method, as OC and EC may come from multiple anthropogenic sources but this method failed to separate them. OC is additionally contributed by secondary process which was also not able to identify by IMB. If performing well, PMF should be able C1to identify the multiple sources for OC and EC, as evidenced in many other places in the world.

Answer: We agree with the referee in relation to the limitations of the IMB method to completely separate and identify all the sources responsible for the carbonaceous matter, because these sources contain other substances besides carbonaceous material. But we continue to disagree with the referee about the capability of the IMB method to provide useful information concerning carbonaceous sources and formation processes. In the present case we did not invest much in separating the non-carbonated, carbonaceous component into fractions, because concentrations are quite low and, for EC, very near the limit of detection of the system analysis capabilities (very low concentrations of EC in a matrix with quite high concentrations of interfering colored dust). In the present case, the total contribution of non-carbonate carbonaceous matter to PM10 is only of the order of 2%. However in another situation where we are applying IMB (concerning urban pollution), where contribution of carbonaceous mater is of the order of 50%, we could separate the carbonaceous matter into biomass burning, primary fossil fuel emission by cars and secondary formation processes. For that, we used levoglucosan measurements, edge lines for EC, versus OC versus fine K, versus levoglucosan and known ratios taken from specialized literature. In this urban data the biomass burning mass contribution compared very well between IMB and PMF (results to be submitted for publication, soon). This is a demonstration of the capabilities of IMB in "more complex" situations.

The fact that the IBM results happen to be consistent with PMF ones is that the PM contributing sources in Cape Verde is quite simple, which cannot sufficiently justify that this method is also applicable in other places with complicated contributing sources. In contrast, PMF has seen successful applications around the world with distinct environmental conditions. Therefore, I consider the approach applied in this study had significant drawback, and the limited scientific significance of this study does not meet the standard of Atmospheric Chemistry and Physics.

Answer: Part of the answer to this commentary has been given in the two previous answers. Here we will only complement the previous exposition. First, we want to stress that we do not have the intention of showing that the IMB should substitute the PMF, neither that it is better. We defend that IMB is a good methodology that extends and deepens frequently used methodologies in the past, which by being able of explaining the total PM mass is therefore more constrained and secured than previous incomplete attempts and that is complementary and useful to evaluate PM sources and formation processes in conjunction with other employed methods, such as PMF. The present evaluated aerosol is simple in what concerns the number of source contributions, taking into account classic air pollution considerations. But it is complex in which concerns collinearity of concentration variations because anthropogenic industrial and transport pollution are originated from the same directions than the sources of dust. Also, dust is emitted with different compositions in different parts of Africa and is important to know the dust contributions from each region. For telling the truth, neither the IMB, nor the PMF, were capable of fully

discriminating the regional origins and contributions of the African dust. As demonstrated in the text now added to the manuscript, in the present situation the PMF could not completely separate even sources such as Sea-salt and Dust, because of the common composition of dust and sea salt, concerning major ions. In many occasions PMF calculated zero sea salt contribution (impossible for a place in the middle of the ocean); in other occasions PMF considered important contributions of Al and Si in the sea salt component. So, this is not a "quite simple" situation.

Anonymous Referee #2

Some improvements have been made in this revised version and I recommend publication in ACP. The paper reported the development of mass balance models for source apportionment of aerosol particles and the models were compared with PMF model. Results showed good agreements between the two models. I have following questions about the ionic mass balance (IMB) model as highlighted below.

(1) Is that possible to compare the mass balanced model developed in the present study with previous works, such as Malm et al., 1994 (JGR, 99, 1347), and other mass balance model thereby to further demonstrate the usefulness of new model.

Answer: The reference in consideration was added to the paper and several sentences were added to the text comparing IMB with previously published Mass Balance attempts.

(2) I would suggest to add some discussions about the robustness of the new IMB model C1as compared with PMF.

Answer: Text is added to the manuscript where the subject is discussed.

(3) Authors mentioned uncertainties in multivariate methods, such as PMF, are there any uncertainties in IMB model developed and applied in this study?

Answer: We have added a subsection in the manuscript where uncertainties for the IMB method are presented and dicussed

Anonymous Referee #3

This paper analyzed elemental and some ion data in PM10 acquired from Cape Verde, an island west of the African Continent, with the attempt to determine the contributions from Saharan dust, sea salt, and their derivatives due to atmospheric transformation. Their approach is mainly based on ion balance, assuming known crustal and sea water chemical composition and the sequence of cation/anion neutralization. The mass and ion closure result from the approach generally make sense, especially when accounting for the residual water content. However, uncertainties were not provided for each contribution estimate, as PMF and other receptor models usually did. In many cases, the authors picked a middle value from a range of possible ratios, such as the water/soluble dust ratio, Fe/Na+ ratio, Mg2+ss/Na+ss, ratio, etc. to carry out their cal culation. Is it possible to propagate uncertainties in these assumptions throughout the calculation and give an overall uncertainty estimate in Table 3? The uncertainties should be compared with those from PMF based on the bootstrapping or DISP methods.

Answer: We have added a subsection in the manuscript where uncertainties for the IMB method are presented and discussed.

In fact, the dust Fe/Na+ ratio (3.7) determined from the edge line (Figure 2) is based on only 3 points and the ratio falls at the low end of possible range of Fe/Na ratio for Sahara dust. It is nearly impossible to find a period with zero sea salt contribution at the island, and therefore using the edge line to determine the dust Na+ fraction is risky. At the very least, uncertainty in this method should be given and propagated into all subsequent calculations.

Answer:

Answer- In this point we do not agree with the referee. This methodology is much better than the known alternative, which is to attribute all $Na^+$ to sea salt production.

In reality the edge line is not based in 3 points only (see Figure).

[Figure]

If we use an estimation taking into account predicable measuring analytical errors, the Fe/Na$^+$ edge ratio is based in 10 points. As important, the edge line intercepts the x axe at Na$^+$ levels within the range of values (represented by the blue rectangle) referring to periods without significant dust intrusions, in accordance with the fact that there is always sea salt spray in this marine atmosphere (The subject is further discussed in the new added sub-section 4.5).

When the authors compare IMB and PMF source contribution, they only compare the average contributions for the three periods. It is also meaningful to compare: 1) correlation of respective source contributions determined by the two methods across individual samples and 2) chemical composition of corresponding sources/classes, particularly the sea salt and dust sources, to examine whether the authors' assumptions in IMB, such as the aforementioned Fe/Na+ ratio, are consistent with PMF. These comparisons should be presented and discussed explicitly.

Answer: New figures with individual comparisons between IMB and PMF (Figure 6a, b, c and d): Comparisons were discussed in added text, showing the capabilities of IMB to source apportion the aerosol.

Additional comments:

Page 10, Line 16. What is the range of this F factor? Can this be propagated into the uncertainty estimate?

Answer: The subject is discussed in new sub-section 4.5.

Page 10, Line 33. Does K+ (and perhaps Cl-) generated from biomass burning need to be considered in the IMB for the region?

Answer: The amounts of possible bio-mass burning material are so small by comparison with other sources that any tentative to detail this source is mostly speculative.

Page 13, Line 12-19. How does the assumption of average inorganic and organic water growth factor influence the mass closure, and the contribution of each source to PM10? Can the range be estimated?

Answer: The subject is discussed in new added 4.5 sub-section.

[revised manuscript text omitted]

---

## Author Response (AR2)

The quality of this manuscript has improved to some extent in the revised version, especially the addition of "Accuracy and Errors" section which is important to alert the readers that both methods have limitations. However, there are still quite a number of questions which hindered me from accepting it for publication. Below are detailed comments.

1.I totally agree that the accuracy of source apportionment can be increased by using different methods, as also suggested by the European Guide. However, I feel that IMB and PMF results, in this case, are not mutually complementary. With respect to PMF, IMB is not totally an "independent methodology". They both use the same input dataset, and essentially attempt to find factors or species that co-vary with time. The edge line methodology is essentially to constrain some factor loadings to be zero in the factor analysis. Therefore, it is quite normal that IMB and PMF results reach a good agreement. Such an agreement is resulted from the similarity in algorithm, which could not provide us additional confidence to the source apportionment results. It would be a completely different story if the results from PMF and CTMs converge or can be improved by their reconciliation, which is the essential meaning of the argument quoted from the European Guide.

**Answer:** We agree that IMB and PMF are not totally independent methods; they use common inference methodologies but put different weight/effort/priority in each of them.

PMF is fundamentally based on the composition relative variability at the receptor to establish a reduction in the number of independent variables to that corresponding to different source or formation processes contributions. Complementary, PMF applies methodologies such as introduction of source profiles to test and constrain the estimated source composions and contributions.

IMB is essentially an ionic and mass balance method, helped by inference/knowledge of the composition of sources or sources classes. IMB uses measured aerosol composition variability (the edge lines) to help in completing source composition information. The results are constrained by equilibrium between cations and anions and the gravimetrically measured particulate mass.

The two methods, by putting a different priority in evaluation mechanisms are therefore somehow complementary. If used together, they increase the degree of confidence in the results (or give an alert in relation to the evaluations if the discrepancies are significant). The present case is a mixture of both outcomes. For some source types, there is a good agreement between IMB and PMF. For others there are relatively important differences that need to be explained and reconciled.

2.In response, the authors claimed that "IMB may help in the obtaining of a more real and accurate solution than that the one obtained only with PMF", which I cannot agree. In this study, the authors admit that IMB cannot discriminate Dust+Ind which PMF can.

**Answer**: It seems that the referee has interpreted that our aim was to demonstrate that the IMB source apportionment technique provides better results than the PMF one, in all the cases. We would like to strongly emphasize that we never intended to defend that IMB is generally superior to PMF, only that it can complement, clarify and, in some cases, correct the results of PMF analysis. In our opinion in the present case, for several of the sources, such as the sea salt spray, the IMB is evidently superior in establishing the true quantitative contribution, than PMF. Of course, even in the present case, there are presumably source evaluations in which PMF is superior to IMB. But we think that "Dust+Ind" is not a good example of PMF superiority. In reality this is not a true simple source, as the proper given name indicates. This is a mixture of two types of sources, industrial emissions and wind-born dust produced in the same trajectory path locations that, by covariating in relative concentrations, could not be clearly separated by PMF.

In addition, PMF can discriminate different sources of carbonaceous species (e.g. combustion, soil…) which IMB cannot. By looking at the results, probably the only drawback of PMF results is the mixing of elements in the sea-salt and combustion sources which are resulted from the overwhelming presence of soil at this specific location, and such a drawback can be overcame by the latest version of PMF which has the function of constraining source profiles by priori information.

**Answer**: As we have already said in the previous round of answers to the referee comments, IMB is also capable of discriminating sources of carbonaceous matter (see the answer to question 4 for further details/clarifications). For example, in relation to the contribution of organic matter from the soil dust, the IMB could easily estimate the aerosol contribution if there exists information about the average composition of North African desert soil in this material (which we could not find in bibliography). But, in Figure 6d, the IMB permits to clarify that there are important contributions of soil dust organic matter that is at the same or higher levels that "Combustion" emissions detected by PMF. We employed the last version of PMF software in our PMF evaluations (and used all available software capabilities) without being able of resolving the problem.

It is not a good practice to selectively emphasize some points while neglecting others to support some pre-designed viewpoints.

**Answer**: In relation to this point we want to emphasize that it is not our intention to forcedly try to demonstrate that IMB is superior to PMF, or even as good. Our emphasis on the qualities of IMB is in part expectable because we are presenting the methodology for publication, but result fundamentally from the criticisms of the referees in relation to presumable (?) weaknesses of IMB. In contesting the referees critics with answers and information that, in our opinion, demonstrate the non-reasonability of most of them, we are off course emphasizing the good points of the IMB method. We want however to attract the referee attention to the fact that we have added an entire subsection to the manuscript

(Section "4.6 Accuracy and Errors") where we expose the main errors and limitations of IMB methodology applied to the present aerosol measurements.

3. In my opinion, IMB is essentially not a 'source' apportionment tool. The core value of doing source apportionment, apart from understanding the atmospheric processes, is more to investigate contributions from different source categories to ambient pollutants so as to make prioritized control strategies to effectively reduce ambient pollution levels. Those identified by IMB, e.g. WetSoilDust ins, WetSIC am, WetCarbon, cannot directly linked with pollution sources, therefore this IMB method has limited utility from a pollution control point of view, especially in some areas with ambient pollution contributed from diverse pollution sources.

**Answer:** Here we do not agree with the referee. We think that the IMB is as much a "source" apportionment tool, as it is PMF. In our answer to point 1 we showed that IMB and PMF use similar principles, differing in the emphasis and priorities. In the present comment the reviewer is not totally coherent because in point 1 he defends "with respect to PMF, IMB is not totally an "independent methodology".

The three examples given by the referee "WetSoilDust ins, WetSIC am, WetCarbon" are completely diverse. "WetSoilDust ins" represents part of a source "Soil dust" and its discrimination is useful for better estimation of sorbed water and, through determination of "WetSoilDust sol" and ionic balance, estimate the importance of secondary reactions between atmospheric acids and soil dust, a considerable fraction of SIC (secondary inorganic compounds) source/formation processes.

"Wet Carbon" is not really a source and, as referred previously, results from a simplified carbon treatment in the present aerosol data, because of the very low levels of carbonaceous matter found. The IMB methodology can however estimate and differentiate aerosol sources and formation processes, when this contamination is relevant (see answer to question 4, for extended clarification).

"WetSIC am" is of course more related with formation processes than with direct emission sources. But this information is quite important to understand the complex pathways between the pollution sources and the impacted receptors. The calculation of "WetSIC" has nothing specially different with the frequently outcomes of other source apportionment methodologies such as PMF. PMF frequently, or always, presents source profiles and contributions that are in reality transformation processes, such is in the present case the "SIC" source. It is interesting that, for that source/formation process class, the IMB and the PMF methods produce relatively well comparable outcomes.

Therefore in our opinion the determinations of these source/process classes have really an important "utility from a pollution control point of view, especially in some areas with ambient pollution contributed from diverse pollution sources".

4. In IMB, all OC and EC are simply grouped together and named "Non-carbonated carbonaceous matter", and the reason for doing so is their low and near-detection-limit concentrations. In comparison, with such low concentrations, PMF can at least identify two factors for OC and EC. This indicates that in both situations, PMF performs better than, at least equal to, IMB, so what's the additional value of IMB in this case?

**Answer:** As said in the answer to this subject in the previous round of comments we did not invest much in separating the non-carbonated, carbonaceous component into fractions, because concentrations are quite low and, for EC, very near the limit of detection of the system analysis capabilities. We also informed that in another situation where we are applying IMB (concerning urban pollution), where contribution of carbonaceous mater is of the order of 50%, we could separate the carbonaceous matter into biomass burning, primary fossil fuel emission by cars and secondary formation processes.

We do not feel that our paper represents a competition between IMB and PMF. As explained in other parts of our answers both methodologies contribute to a better characterization and quantification of aerosol sources and formation processes

The authors further claimed, without any supporting information provided, that in another situation with considerably higher carbonaceous matter, it can be divided into biomass burning, vehicle exhaust and secondary formation process by IMB, and they compare very well with PMF.

**Answer**: As said, we are preparing a manuscript to submit to a scientific Journal, on the subject. However, at this moment, we have already a Conference publication with a summary of the results that illustrate our claims. This publication can be found and downloaded from Research Gate (https://www.researchgate.net/publication/325687496_Source_Apportionment_of_PM25_and_PM10_aerosol_by_Mass_Balance_in_an_urban_atmosphere_during_two_contrasting_seasons).

5. The 13-step calculation procedure is the core part of IMB, and the first step of identifying Fe/Na+ and Fe/Mg2+ edge lines is pivotal and would impact all further calculations. As suggested by the authors, the sampling location is "in the middle of the ocean where it is not expectable to have absence of sea-salt". However, identifying edge lines by the scatterplot essentially assumes that for points along the edge lines, there are little contribution from sea-salt, which is contradictory with the previous argument that sea-salt contribution is ubiquitous.

**Answer:** We agree with the referee that the interpretation of the edge lines would be easier if the complementary source and species (sea spray and $Na^+$ or $Mg^{2+}$, in this case) had periods (samples) without any contribution ($Na^+$ and $Mg^{2+}$ contribution from sea salt equal to zero), which is difficult/impossible to happen in the middle of the Ocean. However, in our opinion, the principle behind the edge line approach does not rely on the fact that there is no contamination, in the edge line area, from one of the sources (in this case the sea spray). What is needed is that, in the samples used to trace the edge line, the concentrations

resulting from the contamination source are at an approximately constant minimum. The constant presence of this source contamination (sea spray) is demonstrated in the paper (Figure 2) by the interceptions of the edge lines with the horizontal axe at $Na^+$ and $Mg^{2+}$ values well above zero.

Therefore, we consider the Fe/Na+ and Fe/Mg2+ values from the edge lines are underestimated to unknown degrees (could be well above 10%), leading to significant uncertainties to the following calculations. The 'real' edge might be way left of the current edge line.

**Answer:** As said in the previous answer the constant presence of sea spray does not hinder the correct determination of edge lines (meaning correct $Fe/Na^+$ and $Fe/Mg^{2+}$ ratios). The use of two edge line ratios decreases the degree of uncertainty. The subsequent application of $Na^+/Mg^{2+}$ sea salt ratio to calculate $Na^+_{ss}$ and, therefore, soil water-soluble ionic concentrations, further reduces error estimation. Contrary to what is suggested by the referee ("The 'real' edge might be way left of the current edge line"), we predict in the manuscript that this three elements methodology will probably result in the somehow underestimation of soil ions common to sea salt.

6. There are two 'Figure 6' and two 'section 4.5' in this revised version.

**Answer:** The error has been corrected in the present version.

**Anonymous Referee #2**

I have no more comments and the paper can be accepted as is

**Anonymous Referee #3**

The authors discussed more on the model uncertainty and comparison with PMF results in the revised manuscript. However, many important findings are not mentioned in the abstract and ACP Discussion Forum. The inconsistency between IMB and PMF are significant for many sources, based on Figure 6 and the authors' explanations, and this proves that the good agreement in the average proportions may just be a coincidence and misleading. It is not fair to say (in the abstract) they: "giving comparable results throughout the sampling campaign. This gives confidence in the capability of both methods, which are complementary, for the source apportionment of aerosol particles." Disagreement and uncertainties/limitations should be mentioned adequately in the abstract.

**Answer:** We agree that the paper abstract and conclusions emphasize excessively the agreement of results between IMB and PMF. As said in previous answers there is a good agreement between both methodologies for the types of sources but there are also important discrepancies for individual samples. Some of the discrepancies result from the different source classification of each method, but several are resultant from different quantifications of source contributions. We introduced changes in the Abstract and Conclusions to better pinpoint the discrepancies between the results of IMB and PMF. The new modified sentences are:

Abstract: *The balance methodology was compared with Positive Matrix Factorization (PMF), showing similar qualitative source composition. In quantitative terms, while for Soil dust and Secondary Inorganic Compounds source classes, the results are similar, for other sources such as sea salt spray there are significant differences in periods of dust episodes. The discrepancies between both approaches are interpreted with basis on calculated source profiles. The joint utilization of the two methodologies, which are complementary, gives confidence in our capability for the correct source apportionment of aerosol particles.*

Conclusions: *The IMB methodology was compared with PMF results applied to the same data set. In seasonal averaged terms the outcomes of the two methodologies were comparable for the most important sources and formation processes. Comparison between individual samples showed however significant differences, principally for the sea-salt spray and the carbonaceous/combustion sources. Because of the overwhelming presence of dust in most samples, the PMF could not clearly separate dust from sea-salt sources. On the other hand, IMB could not discriminate soil organic matter from combustion emissions.*

*We can rely in the complementarity of both methods for the evaluation of sources contributing to atmospheric contamination, in circumstances of very high natural inputs of sea-salt and desert dust particles, subject to atmospheric transformation during long-range transport. Utilization of these two independent source apportionment methodologies adds confidence to the apportionment of an atmospheric aerosol with quite specific and uncommon characteristics*

In addition, many of the authors' responses lack details, which are only presented in the revised manuscript, beating the purpose of having the ACP discussion forum. Key revisions should be mentioned (even by copying and pasting) in the discussion forum. The reviewer does not recommend publication without this being done.

Answer: We apologize for not having introduced in the answers to the previous round of questions all the detail expected and added to the manuscript. Our intention was to not tire the reader with repetitions but we understand that we should have done otherwise, to help in the discussion forum. In the following lines, we try to compensate for the previous failure, repeating the referee previous round of questions and adding a more extended answers text:

Referee 3

This paper analyzed elemental and some ion data in PM10 acquired from Cape Verde, an island west of the African Continent, with the attempt to determine the contributions from Saharan dust, sea salt, and their derivatives due to atmospheric transformation. Their approach is mainly based on ion balance, assuming known crustal and sea water chemical composition and the sequence of cation/anion neutralization. The mass and ion closure result from the approach generally make sense, especially when accounting for the residual water content. However, uncertainties were not provided for each contribution estimate, as PMF and other receptor models usually did. In many cases, the authors picked a middle value from a range of possible ratios, such as the water/soluble dust ratio, $Fe/Na^+$ ratio, $Mg^{2+}ss/Na+ss$, ratio, etc. to carry out their cal culation. Is it possible to propagate uncertainties in these assumptions throughout the calculation and give an overall uncertainty estimate in Table 3? The uncertainties should be compared with those from PMF based on the bootstrapping or DISP methods.

**Answer:** We have added a subsection in the manuscript where uncertainties for the IMB method are presented and discussed. In the new Section "4.5 Accuracy and Errors", besides other statements, we considered that "*in IMB, there are probably four estimations where the errors influencing the source apportionment are higher: a) calculation of water sorbed in sea-salt; b) the estimation of total soil content based in factor F; c) Calculation of sea-salt $Na^+$; d) the estimation of organic matter from OC. It is necessary to take into account however that total errors are controlled by measured $PM_{10}$ total mass constraint (Malm et al., (1994) use this comparison as a validation and self-consistency check for their reconstructed aerosol mass balance)*" We also stated that "*falling the sum of estimated masses within 96-99% of measured $PM_{10}$ total mass, the individual errors are probably limited (or of opposite directions, with mutual compensation)*".

We present in the manuscript new text estimations of the probable errors affecting the four calculations, namely for water-"*Estimation of water content in sea-salt depends mainly on metastable equilibrium considerations that give a water/dry sea-salt mass ratio varying from 0.4 to 1.4. An average value of 0.9 was applied in our calculations. Application of the two extreme values would vary the fractional contribution of wet sea-salt by, approximately, ±6%, ±7% or ±3.5%, for the total sampling campaign, the Dust-Free, or the Dry-Haze periods. It is necessary to consider, however, that by choosing the two ratio extremes the closure of the total PM mass would be affected, resulting in a maximum unaccounted mass of 16%, for the choice of a 0.4 value and a maximum over-prediction of 9%, for the choice of 1.4 ratio and therefore the correctness of these extreme values is questionable*",

For soil dust we considered that- *The calculation of the total soil dust was based in equation 3 that uses an average factor F value of 1.15. F values taken from Moreno et al. (2006) (see Table A1 in annex, for clarification) vary in the range 1.09-1.27 (standard deviation=0.21) and the factor F for global composition is 1.05-1.06. A range of approximately 16% will result from the calculation of soil dust contribution by application*

*of the two extreme F factors. Use of extreme F values would give a maximum unaccounted PM mass of 8% or an excess total PM mass calculation of 6%.*

In the calculation of sea-salt $Na^+$ errors, we introduced in the new text the following discussion: *"The $Fe/Na^+$ edge line methodology used to estimate $Na^+$ and $Mg^{2+}$ was conservative, minimizing the subtraction from sea-salt. This was partially compensated by using minimum values and the $Na^+_{ss}/Mg^{2+}_{ss}$ ratio in seawater. The edge line intercepts the x axe at $Na^+$ levels within the range of values observed for periods without significant dust intrusions, in accordance with the fact that there is always sea salt spray in this marine atmosphere (see Figure 2a, where air masses without dust intrusions are represented by a blue rectangle). The points to the right of the edge line have excess $Na^+$, by comparison with Fe in the edge line. This excess can have two origins: either it results from variability in the relative content of $Na^+$ in soil dust, or it is resultant from the variability in the contributions of sea salt. It is clear that the points at low Fe levels, within the blue rectangle, are only consequence from variability in sea salt spray loading. On the right border of the blue rectangle, a pink straight line is drawn, parallel to the Fe/Ion edge lines. As all the measured points are within both lines, this is a strong indicator that $Na^+$ and $Mg^{2+}$ increase, in relation to the edge lines, result mostly from variability in sea-salt input. The real value of the errors in this methodology is difficult to establish but a change in the $Fe/Na^+$ edge ratio of 10% would have no visible effect in the estimation of sea-salt or soil dust, by IMB*.

In relation to the errors related to calculation of Organic Mater we added the following text: *"The estimated value for non-carbonate carbonaceous matter depends strongly from the OM/OC ratio. In literature values in the range 1.2-2.3 have been proposed. We used a ratio of 2.0 in the high end of the range because the sampling was performed at a background location, away from primary combustion sources, with plenty of time for oxidation and secondary formation. However, during dust episodes, important fractions of the organic material have a soil origin and for that less reliable information exists. As the Carbonaceous fraction in PM mass is only 2-6%, errors in OM/OC ratio have only a small effect in total mass attribution, but important errors, of the order of 40%, can result in the estimation of the Carbonaceous mass if a OM/OC ratio of 1.6 is the correct assumption, instead of 2.0".*

[Figure]

*Figure 2: Edge lines (in red) for Fe versus Na⁺ and Fe versus Mg²⁺ inter-comparisons. The blue rectangle represents periods without significant dust intrusions. The pink lines are parallels to the red edge lines in the maximum ion concentration regions. Also shown Fe/Na ratio ranges in Sahara and global soils, for total sodium, taken from Moreno et al., (2006), and Manson and Moore, (1982).*

In fact, the dust Fe/Na+ ratio (3.7) determined from the edge line (Figure 2) is based on only 3 points and the ratio falls at the low end of possible range of Fe/Na ratio for Sahara dust. It is nearly impossible to find a period with zero sea salt contribution at the island, and therefore using the edge line to determine the dust Na+ fraction is risky. At the very least, uncertainty in this method should be given and propagated into all subsequent calculations.

**Answer-** In this point we do not agree with the referee. This methodology is much better than the known alternative, which is to attribute all Na⁺ to sea salt production.

In reality the edge line is not based in 3 points only (see Figure down).

[Figure]

If we use an estimation taking into account predicable measuring analytical errors, the Fe/Na⁺ edge ratio is based in 10 points. As important, the edge line intercepts the x axe at Na⁺ levels within the range of values (represented by the blue rectangle) referring to periods without significant dust intrusions, in accordance with the fact that there is always sea salt spray in this marine atmosphere

Furthermore for calculation of sea salt ions we use a combination of edge lines for Na+ and Mg²⁺, together with sea-salt Na⁺/Mg²⁺ ratio, which compensates possible individual errors and inaccuracies.

The subject is further discussed in the new added sub-section 4.5 where for Sea-salt estimation, we observed that the comparison between IMB and PMF "*is not so good with IMB/PMF ratio estimation of only 0.68 and an R=0.82. This happens principally because, for several samples, PMF gives zero or negative Sea-salt contributions, while the IMB estimates important Sea-salt values. For a location in the middle of the ocean, it is not expectable to have absence of sea-salt and therefore, in our opinion, PMF fails, attributing,*

*probably, the available sea-salt to a Soil source. At high concentration levels there is a tendency of PMF to give higher sea-salt values than IMB. An inspection of PMF compounds contribution to Sea-salt source shows that, in average, there is a mass inclusion of about 20% of elements such as Si, Al, Fe, etc., in this source (see Figure A8, graph 8, for clarification). Therefore, it is clear that PMF could not completely separate Soil from Sea-salt sources, probably because of the overwhelming presence of soil during dust episodes. During the "Dust-Free" season IMB tends to give somehow higher Sea-salt contributions than PMF (Sea-salt IMB= 0.73 Sea-salt PMF+6.8; R=0.84).  One of the possible reasons may be a too high estimation of sea-salt sorbed water, in IMB."*

When the authors compare IMB and PMF source contribution, they only compare the average contributions for the three periods. It is also meaningful to compare: 1) correlation of respective source contributions determined by the two methods across individual samples and 2) chemical composition of corresponding sources/classes, particularly the sea salt and dust sources, to examine whether the authors' assumptions in IMB, such as the aforementioned Fe/Na+ ratio, are consistent with PMF. These comparisons should be presented and discussed explicitly.

**Answer:** New figures with individual comparisons between IMB and PMF (Figure 6a, b, c and d): Comparisons were discussed in added text, showing the capabilities of IMB to source apportion the aerosol.

[Figure]

*Figure 6: Comparison between the apportionment of individual samples using IMB and PMF, for four source classes. Circles filled in white represent periods with intermittent dust intrusions. Linear best fits are presented for the total sampling campaign.*

Comparisons were discussed in added text, showing the capabilities of IMB to source apportion the aerosol: "*Both methodologies also compare reasonably well in what concerns Secondary Inorganic Compounds (SIC) contributions to the aerosol loading, with a ratio IMB/PMF=0.82 and a correlation coefficient R=0.82. Where the comparison fails*

*completely is in the Carbonaceous (IMB) versus Combustion (PMF) sources that present no clear relation. This results from several facts as exposed in the following text. IMB source profile represents only non-carbonate carbonaceous matter, irrespective of the source, while PMF factor intends to represent all emissions from combustion sources, besides carbonaceous matter. Therefore, from Figure 6 it is possible to observe in several samples important contributions of Carbonaceous matter estimated by IMB, while PMF gives zero to negative Combustion emission estimations. Most probably this results from soil contribution to organic matter that in PMF is attributed to dust or anthropogenic sources (Ant+Dust, see Graphs C and H in Figure A8, for clarification). Also in PMF the Combustion source has, in average, an important contribution (around 50%; see Figure A8, for clarification) of elements, such as Si, Al, Fe, etc., from soil origin and, therefore, in our opinion, this PMF Combustion source is highly contaminated with soil, with PMF not fully capable of separating Combustion from Soil dust, due to the overwhelming presence of soil particles during dust episodes"*

There is a slight difference in Figure 2 in the response and in the revised manuscript as the upper line passes different points (the one in the Response pass more points than the one in the revised manuscript). Why is there such an inconsistency? How do you decide the number of points to be included as "edge"?

**Answer:** As previously said, small variations in the edge line ratios for Fe versus $Na^+$ or $Mg^{2+}$ do not affect appreciably the outcome of sea salt source contribution. In our previous manuscript, we used qualitative visual inspection to trace the edge lines. But for comfort of the readers we decided now to apply a more objective methodology, similar to that already applied in Pio et al. (2011). The technique is based in tracing a linear best fit line through the 5% of total points (7 points in our case) with the highest (Fe/(X-Xmin) ratios, where X is $Na^+$ or $Mg^{2+}$ ion concentration and Xmin is the respective measured minimum ion concentration. As result of this new methodology, there are small variations in the IMB results that are expressed in new figures, tables and text.

In addition, the authors should think about how the various uncertainties can be propagated and presented in Table 3. In the reviewer's opinion, a value without appropriate uncertainty can be misleading.

**Answer:** In the added section "4.6 Accuracy and Errors" we discuss the uncertainties and present possible maximum errors for the presumably four less precise IMB source evaluations. We feel uncomfortable in presenting a precise error figure for each of the calculated sources because we do not have means of an objective evaluation and we think that in these cases a figure can give the reader a false security. Such is the case for the errors calculated by the PMF software that, in our opinion, are underestimated by the software.

Finally, in the conclusion, the authors stated "source composition and contribution knowledge obtained with IMB can be used to constrain a model run with the new versions of the PMF model". Actually this (constraint) can be achieved by the current EPA PMF 5.0 already. You do not need to wait for a "new" version of PMF.

**Answer:** We employed the EPA PMF 5.0 last version in our PMF calculations and we applied in the evaluation all the capabilities of the software, using source constrains. We believe that there has been a misunderstanding in relation with the meaning of "*new versions of the PMF model*". We are aware that the current version of the model offers the option to include source constraints and just wanted to highlight the fact that such constraints can be estimated with IMB. Probably the use of the term "*last versions of the PMF model*" would have been more correct than "*new versions of the PMF model*". We apologize for this and the sentence has been corrected in the reviewed version of the manuscript.

**Source apportionment of atmospheric aerosol in a marine dusty environment by Ionic/composition Mass Balance (IMB)**

João Cardoso[1,2], Susana M. Almeida[3], Teresa Nunes[1], Marina Almeida-Silva[3], Mário Cerqueira[1], Célia Alves[1], Fernando Rocha[4], Paula Chaves[3], Miguel Reis[3], Pedro Salvador[5], Begoña Artiñano[5], Casimiro Pio[1]

[1]CESAM & Dep. Environ, Aveiro University, Aveiro, 3810-193, Portugal
[2]Cape Verde University, Praia, 279, Cape Verde
[3]C²TN, Instituto Superior Técnico, Lisbon University, Bobadela, 2695-066, Portugal.
[4]Geobiotec & Dep. Geosciences, Aveiro University, Aveiro, 3810-193,Portugal
[5]Environ Dep, CIEMAT, Madrid, 28040, Spain

*Correspondence to:* Casimiro Pio (casimiro@ua.pt)

**Abstract.** $PM_{10}$ aerosol was sampled in Santiago, the largest island of Cape Verde, for one year, and analysed for elements, ions and carbonaceous material. Very high levels of dust were measured during the winter months, as a result of the direct transport of dust plumes from the African continent. Ionic and Mass Balances (IMB) were applied to the analysed compounds, permitting the determination of 6-7 different processes and source contributions to the aerosol loading: insoluble and soluble dust, sea-salt, carbonaceous material and secondary inorganic compounds resulting from the reaction of acidic precursors with ammonia, sea-salt and dust. The mass balance could be closed by the consideration and estimation of sorbed water that constituted 20-30% of the aerosol mass. The balance methodology was compared with Positive Matrix Factorization (PMF), giving comparable results showing similar qualitative source composition.throughout the sampling campaign. In quantitative terms, while for source classes such as Soil dust and Secondary Inorganic Compounds source classes, the results are similar, for other sources such as sea-salt spray there are significant differences in periods of dust episodes. The discrepancies between both approaches are interpreted with basis on calculated source profiles with IMB presenting lower average concentration values. The joint utilization of the two methodologies together is 
[revised manuscript text omitted]

---

## Author Response (AR3)

*Accepted as is*

**Anonymous Referee #2**

*Accepted as is*

**Anonymous Referee #3**

*Accepted subjected to minor revisions*

The authors clarified how they determined the Fe/Na+ edge lines (i.e., the 5% rule). Please make sure/clarify if the same rule had been applied to all other edge lines (e.g., those in Figure A4 and A5). If not, why? In some cases, the edge lines can change a lot for different data selection rules.

Answer: In reality, we did not apply the 5% rule to the edge lines estimation in Figures A4 and A5 and this happened because:

-the edge line in Figure A4 is only an illustrative example of an alternative methodology to calculate soil dust, not being applied in any further calculations;

-the edge lines in figure A5 were used for only "a **rough** calculation of the fraction of these ionic compounds that are present in the soil, or that result from secondary reaction with atmospheric produced acids" as explained in point 13 of the ionic balance methodology.

However, taking into account the doubts of the referee and for a uniform methodology, the 5% rule is now applied to all edge lines used in the calculations or in the paper. This resulted in small changes in the IMB estimations that are corrected in new figures and tables.

For the last paragraph, citations should be added for constrained PMF-type approaches in recent literature to alert readers: e.g.:

Chen, L.-W.A. and Cao, J., 2018. PM2.5 Source Apportionment Using a Hybrid Environmental Receptor Model. Environmental Science & Technology, 52(11), pp.6357-6369.

Liu, G.R., Shi, G.L., Tian, Y.Z., Wang, Y.N., Zhang, C.Y. and Feng, Y.C., 2015. Physically constrained source apportionment (PCSA) for polycyclic aromatic hydrocarbon using the Multilinear Engine 2-species ratios (ME2-SR) method. Science of the Total Environment, 502, pp.16-21.

Answer: The proposed references were added to the referred paragraph.

[revised manuscript text omitted]